

# Depth-extrapolation of field-scale soil moisture time series derived with cosmic-ray neutron sensing using the SMAR model

Daniel Rasche[1], Theresa Blume[1], and Andreas Güntner[1,2]

[1]GFZ German Research Centre for Geosciences, Section Hydrology, 14473, Potsdam, Germany
[2]University of Potsdam, Institute of Environmental Sciences and Geography, 14476, Potsdam, Germany

**Correspondence:** Daniel Rasche (daniel.rasche@gfz-potsdam.de)

**Abstract.** Soil moisture measurements at the field-scale are highly beneficial for different hydrological applications including the validation of space-borne soil moisture products, landscape water budgeting or multi-criteria calibration of rainfall-runoff models from field to catchment scale. Many of these applications require information on soil water dynamics in deeper soil layers. Cosmic-ray neutron sensing (CRNS) allows for non-invasive monitoring of field-scale soil moisture across several

hectares around the instrument but only for the first few tens of centimeters of the soil. Simple depth-extrapolation approaches often used in remote sensing applications may be used to estimate soil moisture in deeper layers based on the near-surface soil moisture information. However, most approaches require a site-specific calibration using depth-profiles of in-situ soil moisture data, which are often not available. The physically-based soil moisture analytical relationship SMAR is usually also calibrated to sensor data, but could be applied without calibration if all its parameters were known. However, in particular its water loss

parameter is difficult to estimate. In this paper, we introduce and test a simple modification of the SMAR model to estimate the water loss in the second layer based on soil physical parameters and the surface soil moisture time series. We apply the model at a forest site with sandy soils with and without calibration. Comparing the model results against in-situ reference measurements down to depths of 450 cm shows that the SMAR models both with and without modification do not capture the observed soil moisture dynamics well. The performance of the SMAR models nevertheless meets a previously used benchmark RMSE of

$\leq 0.06 \, \mathrm{cm^3 \, cm^{-3}}$ in both, calibrated and uncalibrated scenarios. Only with effective parameters in a non-physical range, a better model performance could be achieved. Different transfer functions to derive surface soil moisture from CRNS do not translate into markedly different results of the depth-extrapolated soil moisture time series simulated with SMAR. However, a more accurate estimation of the sensitive measurement depth of the CRNS improved the soil moisture estimates in the second layer. Despite the fact that the soil moisture dynamics are not well represented at our study site using physically reasonable

parameters, the modified SMAR model may provide valuable first estimates of soil moisture in a deeper soil layer derived from surface measurements based on stationary and roving CRNS as well as remote sensing products where in-situ data for calibration are not available.



# 1 Introduction

Soil moisture is a key parameter in the hydrological cycle (e.g., Vereecken et al., 2008, 2014; Seneviratne et al., 2010). It
controls several aspects of the environment such as soil infiltration, runoff dynamics, plant growth and biomass production
which in turn influence evapotranspiration as well as the climatic conditions on varying spatio-temporal scales (see reviews
by e.g., Daly and Porporato, 2005; Vereecken et al., 2008; Seneviratne et al., 2010; Wang et al., 2018). Thus, information on
soil water dynamics at the field-scale have great importance for various larger-scale hydrological applications ranging from
landscape water budgeting to multi-criteria calibration approaches in rainfall-runoff modeling. However, due to the high spatio-
temporal variability of soil water content (Famiglietti et al., 2008; Vereecken et al., 2014) which is highest in surface soil layers
(Babaeian et al., 2019), measuring field-scale soil moisture and its dynamics proves difficult based on invasive point-scale soil
moisture measurement methods as for example reviewed in Vereecken et al. (2014) and Babaeian et al. (2019). For instance,
the installation of electromagnetic point sensors measuring at high temporal resolution would require a very large number of
sensors to obtain a representative field-scale average (Babaeian et al., 2019). Additionally, sensor networks are not always
feasible as agricultural management practices hamper a permanent installation of point sensors (Stevanato et al., 2019). As a
consequence, extensive point sensor networks which allow for the estimation of field-scale soil moisture are often restricted
to a rather small number of research related monitoring sites such as the Terrestrial Environmental Observatories (TERENO,
www.tereno.net) in Germany (e.g., Zacharias et al., 2011; Bogena et al., 2018; Kiese et al., 2018; Heinrich et al., 2018).

Kodama et al. (1979), Kodama et al. (1985) and Dorman (2004) suggested the potential of naturally occuring secondary
neutrons produced by high-energy cosmic rays for estimating soil and snow water. About a decade ago Zreda et al. (2008);
Desilets et al. (2010), introduced a methodological framework for soil moisture estimation using cosmic-ray neutrons. The
cosmic-ray neutron sensing (CRNS) approach is a non-invasive geophysical method for estimating representative field-scale
soil moisture (Schrön et al., 2018b) based on the measurement of cosmic-ray neutrons which are inversely related to the amount
of hydrogen in the vicinity of the neutron detector. As soil water is the largest pool of hydrogen in the footprint of the neutron
detector in most terrestrial environments, CRNS allows for the measurement of integrated soil moisture of several hectares
around the instrument and the first decimetres of the soil (e.g., Zreda et al., 2008; Desilets et al., 2010; Köhli et al., 2015;
Schrön et al., 2017).

Estimating soil moisture using CRNS has a high potential for various hydrological applications, which require soil moisture
observations at the field-scale. Several studies demonstrate the potential of CRNS-derived soil moisture estimates for example
for a comparison with satellite derived soil moisture products, their validation and the improved calibration of environmental
models (e.g., Holgate et al., 2016; Montzka et al., 2017; Iwema et al., 2017; Duygu and Akyürek, 2019; Dimitrova-Petrova et al.,
2020). Besides stationary CRNS probes for the retrieval of field scale soil moisture time series, roving CRNS-devices have been
successfully used, mapping CRNS-derived surface soil moisture in even larger areas with instruments mounted on vehicles
(e.g., McJannet et al., 2017; Schrön et al., 2018a; Vather et al., 2019) and (Fersch et al., 2018) illustrate potential synergies
between CRNS, airborne radar and in-situ point sensor networks for soil moisture estimation across spatial scales. Due to the
sensitivity of CRNS to any hydrogen in the measurement footprint, snow monitoring (e.g., Schattan et al., 2017, 2019; Gugerli



et al., 2019), irrigation management (e.g., Li et al., 2019a) as well as biomass estimation (e.g., Baroni and Oswald, 2015; Tian et al., 2016; Jakobi et al., 2018; Vather et al., 2020) pose further fields of application and are reviewed in Andreasen et al. (2017).

Although the large areal footprint of the CRNS-instrument allows estimating field-scale integral soil moisture, the CRNS-derived time series lack soil moisture information from greater depths. However, soil moisture at these greater depths becomes highly relevant as soon as the rooting depth of crops or forest extends past the first decimeters. The maximum rooting depth and hence, root zone extent as well as root density along the soil profile varies with vegetation type and biome (e.g., Canadell et al., 1996; Jackson et al., 1996). According to Jackson et al. (1996), on global average across all biomes, the 75 percent of

plant roots occur in the first 40 centimetres of the soil, which would be largely covered by the CRNS. However, the global average maximum rooting depth, and thus, root zone depth is about 4.6 m (Canadell et al., 1996) where the rooting depth also depends on prevailing soil hydrological conditions (Fan et al., 2017). Even grassy vegetation and crops can have rooting depths of more than 200 cm (Canadell et al., 1996), thus exceeding the measurement depth of CRNS. Deep roots play a significant role for the water supply of plant ecosystems especially during dry conditions (Canadell et al., 1996) i.e. through hydraulic

redistribution (see e.g., Neumann and Cardon, 2012) or increased root water uptake from deeper soil layers under drought conditions (Maysonnave et al., 2022). Furthermore, plant species influence infiltration and vertical soil moisture patterns through species dependent root distributions (e.g. Jost et al., 2012) and horizontal soil moisture patterns through species dependent evapotranspiration and interception rates (e.g. Schume et al., 2003). Hence, field-scale soil water information from the deeper vadose zone overcoming these smaller scale heterogeneities can be important for the quantification of water storage variations,

potential influences on vegetation dynamics, matter fluxes and the characterization of the local hydrological cycle.

Given the importance of soil moisture in the deeper root zone, extending CRNS-measurements to greater depths is of high importance for broadening the applicability of CRNS for soil water estimations (Peterson et al., 2016). Numerous studies extrapolate surface soil moisture time series to greater depths using different empirical approaches (e.g., Zhang et al., 2017; Li and Zhang, 2021) including regression analyses, machine learning techniques or other approaches such as the soil water

index (SWI) (Wagner et al., 1999; Albergel et al., 2008). Few studies address the depth-extrapolation of field-scale CRNS-derived soil moisture time series (e.g., Peterson et al., 2016; Zhu et al., 2017; Nguyen et al., 2019; Franz et al., 2020) to the shallow root zone (approx. 100 cm) by applying and comparing extrapolation approaches with the SWI being the most commonly used approach (e.g., Peterson et al., 2016; Dimitrova-Petrova et al., 2020; Franz et al., 2020). All these approaches require reference soil moisture information in the depth of interest to either build an empirical model or calibrate the depth-

extrapolated soil moisture time series. This information may not always be available in sufficient quantity and quality. In contrast, the physically-based soil moisture analytical relationship (SMAR) (Manfreda et al., 2014), applied and modified in recent studies (e.g., Faridani et al., 2017; Baldwin et al., 2017, 2019; Gheybi et al., 2019; Zhuang et al., 2020; Farokhi et al., 2021), allows for the extrapolation of daily surface soil moisture information to a second, lower soil layer by solely relying on soil physical information and a water loss term. This method does not require calibration if the environmental parameters are

known.



Against this background, we investigate the potential to depth-extrapolate hourly and daily surface soil moisture time series without calibration and thus without the need for reference soil moisture information in the depth of interest by applying the SMAR algorithm at a highly equipped study site in the TERENO-NE observatory located the lowlands of north-eastern Germany. While soil physical parameters may be determined from soil analyses, the water loss parameter describing the water loss per unit time from the second soil layer is more difficult to estimate. Therefore, we propose a simple modification of the SMAR algorithm to estimate the water loss term from soil physical characteristics and from the surface soil moisture time series derived from CRNS. We first compare the standard SMAR that uses a constant, calibrated water loss term (calibrated against in-situ reference sensors) with the modified, uncalibrated SMAR that uses the estimated water term loss for different depths of the second soil layer down to 450 cm depth. Secondly, we calibrate all soil parameters in the original and modified version of the SMAR model in order to assess its best possible performance at the study site for the given in-situ reference data. In addition, we apply different neutron-to-soil moisture transfer functions available to derive the surface soil moisture time series. This is done to assess which transfer function performs best and if a better CRNS-derived surface soil moisture time series translates into better estimates of the depth-extrapolated soil moisture. Lastly, we test the influence of the choice of the depth of first soil layer, i.e. the sensitive measurement depth of CRNS, on the goodness-of-fit of the depth-extrapolated to soil moisture estimates.

## 2 Material and methods

### 2.1 Study site

The study site is located in the TERENO-NE observatory (Heinrich et al., 2018) in the young Pleistocene landscape of north-eastern Germany (Fig. 1). The site hosts the CRNS sensor „Serrahn" (Bogena et al., 2022). The site has a mean annual temperature of 8.8°C and mean annual precipitation of 591 mm per year, measured at the long-term weather station in Waren (in a distance of approximately 35 km) operated by the German Weather Service (station ID: 5349, period 1981–2010) (DWD - German Weather Service, 2020a, b). It is situated on the southern ascent of a glacial terminal moraine formed during the Pomeranian phase of the Weichselian glaciation in the Pleistocene (Börner, 2015). The dominating soil types in the vicinity of the sensor are Cambisols formed on aeolian sands with depths down to 450 cm deposited during the Holocene (Rasche et al., 2023). Continuing downwards, these are followed by deposited glacial till of the terminal moraine, glacio-fluvial sediments and glacial tills originating from earlier glaciations with the latter forming the aquitarde the upper groundwater aquifer with water level depths ranging between 13 and 14 m below the surface (Rasche et al., 2023). A mixed forest dominated by European beech (*Fagus sylvatica*) and Scots pine (*Pinus sylvestris*) is the dominant landcover type. A clearing covered by grassy vegetation can be found nearby.

In order to calibrate the CRNS sensor, soil samples were taken at different distances around the instrument in February 2019 as shown in Fig. 1. Soil samples were taken in 5 cm depth increments from 0–35 cm using a split tube sampler containing sampling rings in order to derive soil moisture, soil physical characteristics, average grain size distributions, soil organic matter and lattice water from laboratory analyses as shown in Tab. 1. Soil moisture and soil bulk density were determined from oven-



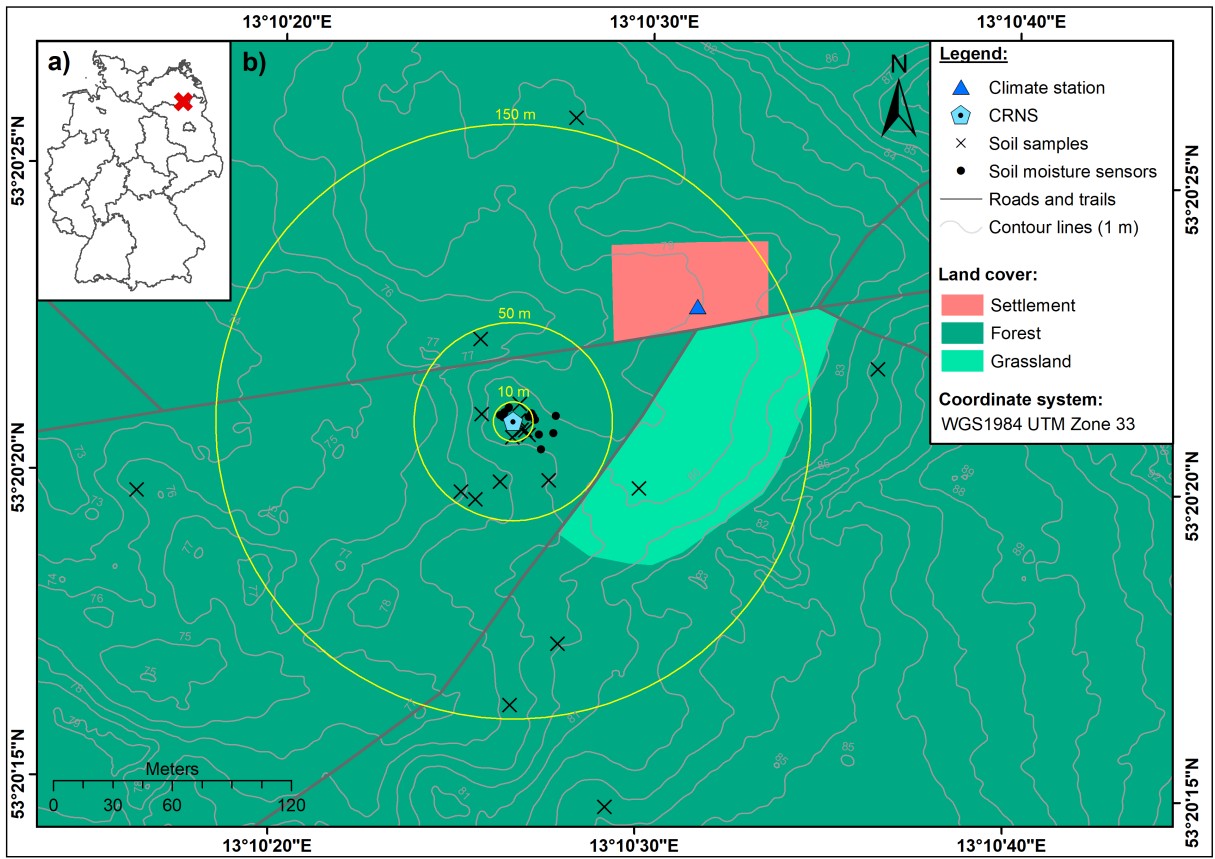

**Figure 1.** Location of the study area within Germany (a) and location of the CRNS observation site „Serrahn" (b) (digital elevation model: LAIV-MV - State Agency for Interior Administration Mecklenburg-Western Pomerania (2011), land cover: BKG - German Federal Agency for Cartography and Geodesy (2018)).

drying at 105°C for 12 h and gravimetric analyses of all individual soil samples. Subsequent loss-on-ignition analyses at 550 and 1000°C with a duration of 24 h were used to determine the amount of soil organic matter and lattice water from bulk samples per depth assuming that no inorganic carbon is present in the acidic aeolian sands. Soil porosity was estimated based on the material density of quartz ($2.65\,\mathrm{g\,cm^{-3}}$) and corrected for the amount of soil organic matter based on the density of cellulose ($1.5\,\mathrm{g\,cm^{-3}}$).

In addition to the stationary CRNS instrument, the study site is equipped with a groundwater observation well, a weather station and a network of in-situ point-scale soil moisture sensor profiles (type SMT100; Truebner GmbH, Germany). A total of 59 in-situ soil moisture sensors is deployed in depths down to 450 cm depth with 12 sensors in 10 cm, 6 sensors in 20 cm, 8 sensors in 30 cm, 8 sensors in 50 cm, 6 sensors in 70 cm, 4 sensors in 130 cm, 7 sensors in 200 cm, 4 sensors in 300 cm as well





as 450 cm. The sensors are located in distances up to 22 m from the CRNS instrument and continuously monitor the volumetric soil moisture content based on the manufacturer's calibration function.

**Table 1.** Soil physical characteristics at the CRNS site Serrahn obtained from laboratory analyses of soil samples (Rasche et al., 2023, modified). Below the maximum sampling depth of 35 cm and down to the maximum depth of the aeolian sand deposits, the soil physical are assumed to have the same soil physical parameters as the layer between 30 and 35 cm. The soil moisture content at field capacity and wilting point were taken from tabulated values in Sponagel et al. (2005) according to the respecitve soil grain size class (medium-fine sand) and the soil bulk density of the individual layers.

| Depth | Grain size fractions | | | | | Bulk density | Porosity | Organic matter | Lattice water | Field capacity | Wilting point |
|---|---|---|---|---|---|---|---|---|---|---|---|
| [cm] | [weight-%] | | | | | [g cm$^{-3}$] | [-] | [g g$^{-1}$] | [g g$^{-1}$] | [cm$^3$ cm$^{-3}$] | [cm$^3$ cm$^{-3}$] |
| | > 2 mm | 2 - 0.63 mm | 0.63 - 0.2 mm | 0.2 - 0.063 mm | < 0.063 mm | | | | | | |
| 0–5 | 2.7 | 19.7 | 42.2 | 33.7 | 2.1 | 0.24 | 0.91 | 0.32 | 0.003 | 0.16 | 0.06 |
| 5–10 | 1.1 | 8.7 | 43.5 | 45.7 | 2.4 | 0.77 | 0.70 | 0.10 | 0.002 | 0.16 | 0.06 |
| 10–15 | 0.7 | 7.2 | 41.5 | 47.9 | 2.8 | 1.25 | 0.52 | 0.05 | 0.002 | 0.16 | 0.06 |
| 15–20 | 1.2 | 7.8 | 38.7 | 44.3 | 2.2 | 1.43 | 0.45 | 0.02 | 0.002 | 0.14 | 0.05 |
| 20–25 | 1.7 | 7.7 | 42.2 | 46.5 | 2.2 | 1.55 | 0.41 | 0.02 | 0.002 | 0.14 | 0.05 |
| 25–30 | 1.7 | 8.5 | 43.5 | 45.4 | 1.2 | 1.59 | 0.40 | 0.01 | 0.002 | 0.12 | 0.04 |
| 30–35 | 1.1 | 8.0 | 42.8 | 46.8 | 1.5 | 1.63 | 0.38 | 0.01 | 0.002 | 0.12 | 0.04 |
| 35–450 | 1.1 | 8.0 | 42.8 | 46.8 | 1.5 | 1.63 | 0.38 | 0.01 | 0.002 | 0.12 | 0.04 |

## 2.2 Field-scale surface soil moisture derived with CRNS

Secondary neutrons are produced by primary cosmic-rays interacting with matter in the atmosphere and in the ground. Depending on their energy level, secondary neutrons may be classified as fast (0.1-10 MeV), epithermal (> 0.25-100 keV) and thermal neutrons (< 0.25 eV) (e.g., Köhli et al., 2015; Weimar et al., 2020). Cosmic-ray neutron sensing for soil moisture estimation relies on the amount of neutrons in the epithermal energy range produced by nuclear evaporation in the atmosphere and ground (Köhli et al., 2015). Epithermal neutrons are sensitive to elastic scattering by collision with hydrogen and are further moderated to thermal neutrons (< 0.25 eV). Thus, the amount of epithermal neutrons detected by the instrument is inversely correlated with the amount of hydrogen in the sensitive measurement footprint of the sensor.

Epithermal neutron counts detected by the instrument are influenced by atmospheric pressure, the amount of primary high-energy cosmic-ray neutrons entering the earth's atmosphere from space (Zreda et al., 2012) as well as variations of absolute air humidity (Rosolem et al., 2013) and need to be corrected for these influencing factors before soil moisture information can be derived. In this study, we use the correction procedure for air pressure and incoming primary cosmic-ray flux presented in Zreda et al. (2012). The correction factor for the shielding effect of the atmosphere can be calculated from local air pressure measurements where the attenuation length $L$ is set to 135.9 g cm$^{-2}$ for the study area (Heidbüchel et al., 2016). The correction factor for the incoming high-energy primary neutron flux was obtained from hourly pressure and efficiency corrected primary neutron intensities (cps) of the Jungfraujoch neutron monitor (JUNG, www.nmdb.eu). Furthermore, the neutron data was corrected for the influence of absolute air humidity introduced by Rosolem et al. (2013). The absolute humidity is calculated



from relative humidity and temperature observations of the weather station at the observation site according to Rosolem et al. (2013). For all correction approaches, the time series averages of air pressure, incoming radiation and air humidity are used as the required reference values. Finally, a 25 h moving average filter is applied to the corrected neutron time series to reduce
noise and uncertainty in the data (e.g., Schrön et al., 2018b).

$$\theta_{\text{Standard}} = \left( \left( \tilde{a_0} \, \frac{1 - \frac{N_{\text{pih}}}{N_{\text{max}}}}{\tilde{a_1} - \frac{N_{\text{pih}}}{N_{\text{max}}}} \right) \times \frac{\rho_{\text{soil}}}{\rho_{\text{water}}} \right) - (\theta_{\text{SOM}} + \theta_{\text{LW}}) , \tag{1}$$

where

$$\tilde{a_0} = -a_2 , \tag{2}$$

$$\tilde{a_1} = \frac{a_1 a_2}{a_0 + a_1 a_2} , \tag{3}$$

$$N_{\text{max}} = N_0 \cdot \frac{a_0 + a_1 a_2}{a_2} . \tag{4}$$

Desilets et al. (2010) introduced a transfer function to convert neutron counts into soil moisture by calibration against reference measurements. Although other approaches exist (e.g., Franz et al., 2013; Köhli et al., 2021), the Desilet's equation became the methodological standard and can be rewritten as eq. (1) – (4) (Köhli et al., 2021) with $a_0 = 0.0808$, $a_1 = 0.372$, $a_2 = 0.115$ and $N_0$ being a local calibration parameter describing the neutron intensity above dry soil (Desilets et al., 2010).
As observed epithermal neutron intensities are sensitive to any hydrogen present in the measurement footprint, the water equivalent of soil organic matter $\theta_{\text{SOM}}$ and the amount of lattice water $\theta_{\text{LW}}$ in $\text{cm}^3 \, \text{cm}^{-3}$ need to be subtracted. Additionally, $\rho_{\text{soil}}$ describes the average soil bulk density in the measurement footprint ($\text{g cm}^{-3}$) and $\rho_{\text{water}}$ the density of water assumed to be 1 $\text{g cm}^{-3}$. In this neutron-to-soil moisture transfer function the neutron intensity corrected for variations in air pressure, incoming primary neutron flux and absolute humidity $N_{pih}$ is used. However, the more recent study by Köhli et al. (2021) suggests that
the influence of absolute air humidity and soil moisture on the observed epithermal neutron signal are interdependent, i.e. the shape of the neutron-soil moisture relationship changes with absolute humidity. The universal transport solution (UTS), eq. (5) – eq. (6), (Köhli et al., 2021) accounts for the changing relationship between neutrons and soil moisture under different conditions of absolute humidity.

$$N_{\text{pi}} = N_D \cdot \left( \frac{p_1 + p_2 \, \theta_{\text{total}}}{p_1 + \theta_{\text{total}}} \cdot \left( p_3 + p_4 \, h + p_5 \, h^2 \right) + e^{-p_6 \, \theta_{\text{total}}} \left( p_7 + p_8 \, h \right) \right) , \tag{5}$$

where



$$\theta_{\text{total}} = \left(\theta_{\text{UTS}} + \theta_{\text{SOM}} + \theta_{\text{LW}}\right) \cdot \frac{1.43\,\text{g\,cm}^{-3}}{\rho_{\text{soil}}} \tag{6}$$

The UTS is designed to describe the neutron intensity response caused by changes in total soil water content and absolute air humidity and therefore, the predicted neutron intensity represents the intensity corrected for variations in atmospheric pressure and incoming primary neutron flux $N_{pi}$. Soil moisture can be derived from the UTS using numerical inversion or a look-up table approach which is used in this study. Analogously to the standard transfer function, the UTS needs to be calibrated locally. The calibration parameter $N_D$ may be interpreted as the average neutron intensity of the local neutron detector under the boundary conditions defined in the neutron transport simulations which where used to subsequently derive the UTS. $\theta_{\text{total}}$ describes the total water content comprising the sum of all below-ground hydrogen pools, namely the soil moisture content $\theta_{\text{UTS}}$, $\theta_{\text{SM}}$ and $\theta_{\text{LW}}$ which is then scaled by ratio of the soil bulk used in the neutron transport simulations to derive the UTS ($1.43\,\text{g\,cm}^{-3}$) and the local soil bulk density at the study site $\rho_{\text{soil}}$ (Köhli et al., 2021). Different sets of shape-giving parameters $p_1 - p_{10}$ are available for the UTS in Köhli et al. (2021) and originate from the different neutron transport models used and whether a simple energy window threshold (thl) was used (parameter sets: URANOS thl, MCNP thl) to evaluate the neutron transport simulations or a more complex detector response function was applied (parameter sets: URANOS drf, MCNP drf). The latter mimics the response of a real neutron detector and is therefore expected to provide more accurate results. In the scope of this study, we investigate which of the two transfer functions and which parameter set of the UTS performs best in estimating surface soil moisture.

The CRNS footprint diameter as well as the integration depth decrease with i.e. increasing soil water content. The radius ranges between 130 and 240 m and the integration depth ranges between 15 and 83 cm during wet and dry conditions, respectively (Köhli et al., 2015). In addition, further factors may influence the footprint dimensions such as open water or topography (e.g., Köhli et al., 2015; Schattan et al., 2019; Mares et al., 2020). Consequently, reference measurements need to be depth-distance weighted according to the sensitivity of the CRNS instrument in order to match field observations of reference measurements when calibrating the two different transfer functions and derive soil moisture information from observed neutron intensities. In this study, we adapt the weighting procedure proposed by Schrön et al. (2017) which takes the total water content, average bulk density, absolute air humidity and vegetation height (set to 20 m) into account. Reference soil moisture information from the soil sampling campaign in February 2019 was weighted accordingly and used for calibrating both transfer functions. In a second step, the CRNS-derived soil moisture time series are compared to an analogously weighted average of all available in-situ soil moisture sensors in 10, 20 and 30 cm depth. In order to assess the impact of weighting procedure, the calibration is repeated using the arithmetic soil moisture average from soil samples and comparing the CRNS-derived soil moisture time series to the arithmetic average soil moisture time series from in-situ sensors.





## 2.3 Depth-extrapolation of surface soil moisture time series from CRNS

### 2.3.1 Modification of the SMAR model

To estimate depth-extrapolated soil moisture time series for a second, deeper soil layer from CRNS-derived surface soil moisture time series, the SMAR model is used. Introduced by Manfreda et al. (2014), it allows for the physically-based estimation of soil moisture in an adjacent second, lower soil layer from soil moisture information in a first, upper soil layer. SMAR is based on the relative saturation in the first and second layer $s_1$ (-) and $s_2$ (-), respectively, the relative saturation at field capacity $sc_1$ (-) and wilting point $sw_2$ (-). In order to transform values from $\mathrm{cm}^3\,\mathrm{cm}^{-3}$ to relative saturation, the respective variables are divided by the porosity of the individual layer $n_1$ ($\mathrm{cm}^3\,\mathrm{cm}^{-3}$) and $n_2$ ($\mathrm{cm}^3\,\mathrm{cm}^{-3}$). After applying the SMAR model, the resulting relative saturation time series of the second layer $s_2$ (-) is transformed back to volumetric soil moisture in $\mathrm{cm}^3\,\mathrm{cm}^{-3}$ by multiplication with $n_2$ ($\mathrm{cm}^3\,\mathrm{cm}^{-3}$) and resulting in the depth-extrapolated soil moisture time series $\theta_{Layer\,2}$. Soil moisture in layer 2 at time $t$ is calculated with

$$s_2\left(t_i\right) = s_{\mathrm{w}2} + \left(s_2\left(t_{i-1}\right) - s_{\mathrm{w}2}\right) \cdot e^{-a \cdot \left(t_i - t_{i-1}\right)} + \left(1 - s_{\mathrm{w}2}\right) \cdot b \cdot y\left(t_i\right) \cdot \left(t_i - t_{i-1}\right), \tag{7}$$

where $a$ and $b$ depend on the vertical extent of the first layer ($Zr_1$ in mm) which begins at the soil surface, and the vertical extent of the second layer ($Zr_2$ in mm). $Zr_2$ is the difference between the maximum depth of the second soil layer and $Zr_1$. The water loss term $V_2$ ($\mathrm{mm\,t}^{-1}$) comprises the bulk water losses from the second layer due to percolation and evapotranspiration per unit time:

$$a = \frac{V_2}{\left(1 - s_{\mathrm{w}2}\right) \cdot n_2 \cdot Zr_2}, \tag{8}$$

$$b = \frac{n_1 \cdot Zr_1}{\left(1 - s_{\mathrm{w}2}\right) \cdot n_2 \cdot Zr_2}, \tag{9}$$

The fraction of saturation of the first layer that instantaneously infiltrates into the second layer $y(t_i)$ (-) is described as (e.g., Manfreda et al., 2014; Patil and Ramsankaran, 2018):

$$y\left(t_i\right) = \begin{cases} \left(s_1\left(t_i\right) - s_{\mathrm{c}1}\right), & s_1\left(t_i\right) \geq s_{\mathrm{c}1} \\ 0, & s_1\left(t_i\right) < s_{\mathrm{c}1}. \end{cases} \tag{10}$$

The SMAR model can be applied using known soil physical and environmental variables. However, although the soil physical parameters may be estimated through pedotransferfunctions, using tabulated values or global soil databases (e.g. SoilGrids 2.0 (Poggio et al., 2021)), the bulk water loss from the second layer $V_2$ is more difficult to estimate. This hampers the use of SMAR





without calibration against reference soil moisture information in the depth of interest, i.e., in the deeper soil layer. To overcome

this issue we modified and extended the SMAR model in order to estimate the $V_2$ based on simple soil physical, environmental

variables and the surface soil moisture time series. A modification of the SMAR model with an extended definition of the water

loss term $V_2$ has been suggested by Faridani et al. (2017) leading to an improved performance compared to the original SMAR

model. As any modification makes the SMAR model more complex and potentially less easy to apply, our aim was to keep the

added complexity to the model low by only including 3 additional parameters. These are the relative saturation at field capacity

in the second layer $sc_2$ (-) and the cumulative root fraction to the maximum depth of the first and second layer $R_1$ (-) and $R_2$

(-), respectively. The water loss term is then defined as the sum of evapotranspiration $ET_2$ (mm t$^{-1}$) and percolation $P_2$ (mm

t$^{-1}$) from the second layer.

$$V_2 = ET_2 + P_2, \tag{11}$$

We adapt the suggestion of Manfreda et al. (2014) to make use of existing (surface) soil moisture time series to gain infor-

mation about water loss from the soil by evapotranspiration at a study site. Here, we estimate the amount of evapotranspiration

from the deeper layer $ET_2$ based on the difference between the current and past value of relative saturation of the first layer,

by scaling the value to the dimension (i.e. extent) of second layer and by considering the difference in cumulative root fraction

between both layers, assuming that root water uptake for ET is larger in the layer with more roots eq. (13). The required root

fraction $R$ (-) for maximum depth $d$ (cm) of the first and second layer are derived from the empirical equation (eq. 12) for

forest biomes presented in Jackson et al. (1996):

$$R = 1 - 0.970^d \tag{12}$$

Using eq. 13, $ET_2$ can only be estimated from the change in relative saturation in the first layer when 1) the relative saturation

of the first layer $s_1$ decreases, 2) no infiltration into the second layer occurs and 3) the relative saturation of the second layer

exceeds the relative saturation at wilting point. This means that both, surface evaporation and transpiration losses are scaled

from the first layer to the second layer. Although surface evaporation is hardly relevant for the second layer due to its missing

connection with the surface, this is a reasonable yet simplified approach because surface evaporation is a comparatively small

component of total evapotranspiration in forests, with transpiration dominating ET (e.g., Li et al., 2019b; Paul-Limoges et al.,

2020).

$$ET_2(t_i) = \begin{cases} (s_1(t_i - 1) - s_1(t_i)) \cdot n_1 \cdot Zr_1 \cdot \frac{Zr_2}{Zr_1} \cdot \frac{(R_2 - R_1)}{R_1}, & s_1(t_i - 1) \geq s_1(t_i); \, y(t_i) > 0; \, s_2(t_i - 1) \leq s_{w2} \\ 0, & otherwise. \end{cases} \tag{13}$$

The amount of percolation $P_2$ from the second layer is estimated in analogy to the infiltration into this layer as an instanta-

neous water loss when the relative saturation exceeds field capacity $sc_2$ (eq. 14).



$$P_2\left(t_i\right)=\begin{cases}\left(s_2\left(t_i-1\right)-s_{\mathrm{c2}}\right), & s_2\left(t_i-1\right)\geq s_{\mathrm{c2}}\\0, & s_2\left(t_i-1\right)<s_{\mathrm{c2}}.\end{cases}\tag{14}$$

### 260  2.3.2  Application of the SMAR model

We applied the SMAR model in its original form by calibrating the $V_2$ water loss term as a constant value. The calibration and evaluation was performed against an average soil moisture time series in the deeper layer derived from in-situ soil moisture sensors. All available in-situ sensor soil moisture time series per depth were averaged to derive average soil moisture time series per sensor depth. Subsequently, we calculated an average soil moisture time series for the second, deeper soil layer by

weighting the averages per depth according to their representative layer extent (called reference time series in the following). For example, having soil moisture sensors installed in 30, 50 and 70 cm depth, the average soil moisture content per time step of all sensors installed in 50 cm is representative for the layer between 40 and 60 cm. The soil physical parameters assigned to the individual layers can be found in Tab, 1. The calibration is performed by minimising the root-mean square error (RMSE) between the depth-extrapolated soil moisture time series and the entire reference soil moisture time series in the second soil

layer.

The original SMAR with calibrated $V_2$ and the modified SMAR model with estimated $V_2$ are applied to estimate a soil moisture time series in a second soil layer with a maximum depth below terrain surface of 70, 130, 200, 300 and 450 cm. In contrast, the modified SMAR model based on eq. 11–14 is applied using the same soil physical parameters but it does not require calibration of the $V_2$ water loss term.

In order to test if a better surface soil moisture time series translates to better extrapolated soil moisture values in the second layer, we apply the SMAR model using the CRNS-derived surface soil moisture estimated from the standard transfer function (eq. (1)) as well as using the UTS (eq. (5) – (6)) with the parameter set resulting in the highest goodness-of-fit expressed by the lowest RMSE.

The vertical extent of the first soil layer is defined according to the representative measurement depth of the CRNS-derived

soil moisture time series. In first step, the model is tested using a depth of the first layer of 35 cm as the sensitive measurement depth of CRNS is often estimated to range between 30 and 40 cm. However, more accurate approaches exist to determine the sensitive measurement depth. In this study, we also calculate median CRNS measurement depth of the entire CRNS-soil moisture time series based on Schrön et al. (2017) and use it as the depth of the first soil layer in the SMAR model. According to Schrön et al. (2017), the sensitive measurement depth $D_{86}$ is estimated using the calibrated CRNS-derived soil moisture

time series for distances from 1 to 300 m around the instrument. Subsequent averaging allows for estimating the average measurement depth in the CRNS footprint for each time step of the time series. The time series median measurement depth $D_{86}$ is then calculated for the soil moisture time series derived with the standard transfer function and the UTS. For both CRNS-derived soil moisture time series, the estimated median sensitive measurement depth is 20 cm and much smaller than



the rough initial estimate of 35 cm. As a consequence, we decided to apply the original and modified SMAR model with a first
layer depth of both 20 cm and 35 cm to investigate the effect on the resulting depth-extrapolated soil moisture time series.

In summary, for each maximum depth of the second soil layer, the SMAR model is applied in its original form based on
calibration and in the modified version presented in this study which does not require calibration. This is done using the CRNS
surface soil moisture time series based on the standard transfer function as well as on the UTS. Lastly, we test whether the
estimation of the representative measurement depth of CRNS and thus, the depth of the first soil layer, has an influence on the
resulting modelled soil moisture time series in the second layer. An overview of the different applications of the SMAR model
performed in this study is given in Fig. 2.

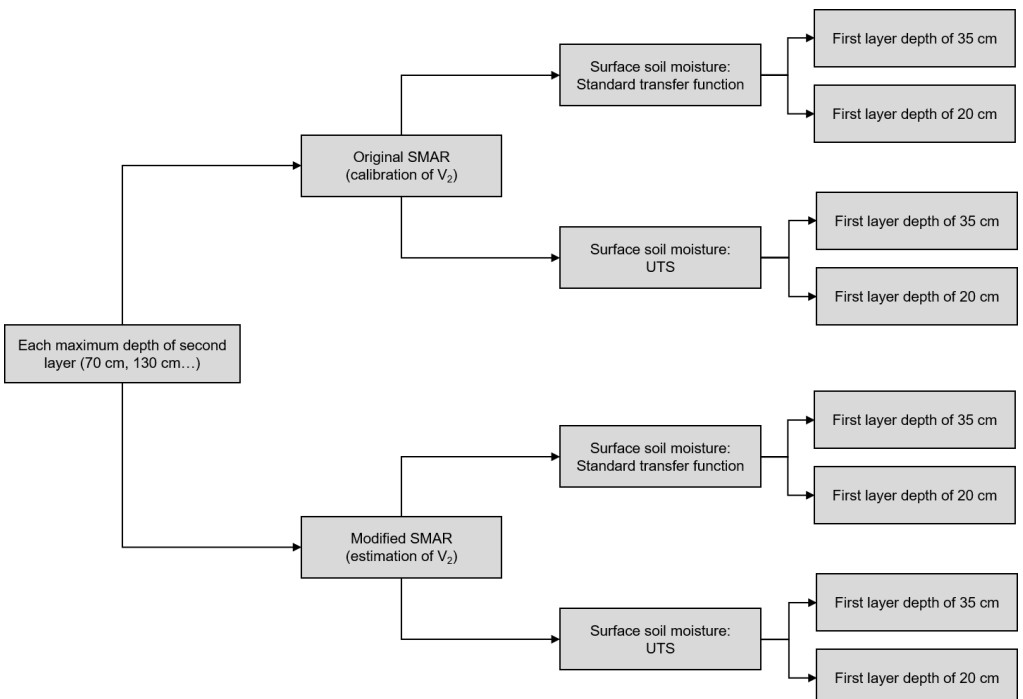

**Figure 2.** Overview of SMAR models set up in the scope of this study to compare the original SMAR based on the calibration of the water
loss $V_2$ and the modified SMAR which does not require calibration.

To assess the robustness of the modified and uncalibrated SMAR model, we made a simple assessment of parameter un-
certainty and its effect on the model results. We set up an ensemble of 50 realizations of the modified SMAR by randomly
varying the values for $n_1$, $n_2$, $sc_1$, $sc_2$, $sw_2$ and $R_1$ in a range of $\pm\,10\,\%$. These sensitivity runs of the modified, uncalibrated
SMAR are then assessed using the minimum to maximum range of calculated root mean square error (RMSE) between the
simulated soil moisture time series in the second soil layer and the reference from in-situ sensors. In satellite soil moisture
estimations, a threshold of 0.06 cm$^3$ cm$^{-3}$ has been used (Jackson et al., 2010) to evaluate the original SMAR performance for





root-zone soil moisture estimates based on satellite-derived surface soil moisture information (Baldwin et al., 2019). We adopt this benchmark for evaluating the performance of the uncalibrated, modified SMAR in this study.

Lastly, we performed a full calibration of the original and modified SMAR models in order to estimate the best possible simulation results within a physically acceptable range of the model parameters. This was done by randomly varying the values for $n_1$, $n_2$, $sc_1$, $sc_2$, $sw_2$ and $R_1$ in a range of $\pm\ 20\,\%$. For the original SMAR model, the water loss term $V_2$ was calibrated instead of the $R_1$ with values in the range between 1 and a maximum of 500 mm. The calibration was performed by selecting the parameter combination that resulted in the lowest RMSE among a total of 10,000 random parameter sets. For

the full calibration scenarios, we defined the year 2017 as the calibration period, while the entire study period (2016-2022) is used to evaluate the depth-extrapolated soil moisture time series. Except for the fact that several parameters were calibrated, the different scenarios are identical to those where just the $V_2$ parameter was calibrated (see Fig. 2). In all SMAR applications in this study, the initial soil moisture content of the second layer was set to the first CRNS-derived soil moisture record of the first layer. The SMAR model with physically reasonable environmental value ranges showed generally low performance

as described in the results section and led to some additional calibration tests of the model. Experimental calibration runs indicated that calibration parameters in a non-physical value range could produce better model results. Therefore, a second full calibration was performed where the values of the parameters $sc_1$, $sc_2$ and $sw_2$ where allowed to range from -1 to their initial, literature-based value while the range for other model parameters remained unchanged.

    The SMAR model was originally designed to depth-extrapolate surface soil moisture time series on a daily resolution, as

it assumes that all water above field capacity $sc_1$ infiltrates into the second layer within one day (Manfreda et al., 2014). Consequently, the SMAR model has been applied on a daily resolution in previous studies (e.g., Baldwin et al., 2017, 2019). In CRNS research, an hourly temporal resolution is the community standard and therefore, we test whether the SMAR models in their original and modified forms can also be applied at hourly resolution with a reasonable goodness-of-fit. All analyses described in this chapter are therefore carried out on both a daily and hourly basis.

All calculations were performed in R statistical software (R Core Team, 2018, 2023) using the hydroGOF package (Zambrano-Bigiarini, 2017, 2020) for calculating goodness-of-fit measures which evaluate absolute values and time series dynamics, namely the RMSE, the Kling-Gupta-Efficiency (KGE) (Gupta et al., 2009) as well as the Pearson correlation coefficient.

## 3   Results and discussion

### 3.1   CRNS-derived surface soil moisture time series

The goodness-of-fit of the calibrated CRNS-based soil moisture time series to the time series derived from in-situ point observations is shown for the two transfer functions Tab. 2. When the different transfer functions are calibrated against an arithmetic average soil moisture from soil samples and compared to an arithmetic average of soil moisture time series in 10-30 cm depth, the Pearson correlation coefficient and the KGE are lower than when using a weighted average of soil moisture observations for calibration as proposed by Köhli et al. (2015) and Schrön et al. (2017). However, the RMSE is slightly higher for the calibration

against the weighted observations. This might be linked to differences between the laboratory measurements of soil moisture



in the soil samples (which were used for calibration) and the continuous soil moisture data obtained from the in-situ sensors. Overall, however, in view of the much better KGE and correlation values, the results underline the importance of the weighting procedures when calibrating the CRNS observations to derive soil moisture estimates or comparing them to observations from in-situ soil moisture sensors.

**Table 2.** Goodness-of-fit between the CRNS-derived soil moisture time series and the arithmetic and weighted average soil moisture time series from the local in-situ point-sale soil moisture sensors in 10-30 cm depth. The different neutron to soil moisture transfer functions are independently calibrated against soil moisture from soil samples taken in February 2019. The UTS transfer function can be used with different parameter sets originating from different neutron transport models which are either based on an energy level threshold (thl) or a more realistic detector response functions (drf).

| Transfer function | In-situ soil moisture | Calibration parameter [cph] | KGE [-] | RMSE [cm$^3$ cm$^{-3}$] | Pearson correlation [-] |
|---|---|---|---|---|---|
| Revised standard | | 777 | 0.08 | 0.030 | 0.88 |
| UTS URANOS drf | | 1245 | 0.14 | 0.029 | 0.86 |
| UTS URANOS thl | Arithmetic average | 1596 | 0.59 | 0.020 | 0.87 |
| UTS MCNP drf | | 1294 | 0.33 | 0.025 | 0.87 |
| UTS MCNP thl | | 1645 | 0.59 | 0.021 | 0.87 |
| Revised standard | | 809 | 0.46 | 0.030 | 0.91 |
| UTS URANOS drf | | 1302 | 0.49 | 0.029 | 0.89 |
| UTS URANOS thl | Weighted average | 1693 | 0.81 | 0.022 | 0.90 |
| UTS MCNP drf | | 1357 | 0.60 | 0.027 | 0.90 |
| UTS MCNP thl | | 1741 | 0.77 | 0.023 | 0.90 |

The goodness-of-fit of the CRNS-derived soil moisture time series that are based on the revised standard transfer function is always lower than for those that are derived with the UTS all parameters sets, especially when the KGE is considered, showing the improved soil moisture estimation with the UTS. However, the parameters sets of the UTS mimicking the varying sensitivity of a real neutron detector to neutrons of different energies (URANOS drf, MCNP drf) perform worse than those which rely on a simple energy range threshold (URANOS thl, MCNP thl). This counter-intuitive result has been previously described by

Köhli et al. (2021) and could be related to the high sensitivity of the CRNS method to the soil moisture dynamics in the first few centimetres of the soil where unfortunately no in-situ sensors are installed (the uppermost sensors are installed in 10 cm depth). Therefore, the better performance of the energy threshold parameters sets of the UTS can be related to insufficient reference soil moisture information from the in-situ sensor network. Generally, the UTS with the parameter sets representing the response of a real neutron detector can be expected to be provide more accurate results. Here, the UTS with parameter set

MNCP drf reveals a higher statistical goodness-of-fit compared to the URANOS drf parameter set which is in line with the findings presented in Köhli et al. (2021). The improved performance of the UTS with the parameter set MNCP drf compared



to the standard transfer function is shown in Fig. 3, revealing that the latter tends to overestimate soil moisture under the wet winter conditions and underestimate soil moisture under dry summer conditions.

Different from the study of Köhli et al. (2021) which introduced the UTS, we apply UTS to derive soil moisture from neutron observations at a forest site. The UTS calibration parameter $N_D$ represents the average count rate under boundary conditions of the neutron transport simulations conducted to the derive the UTS. Therefore, $N_D$ can be expected to be close to the average corrected neutron intensity observed at a study site with little or without vegetation or other above-ground hydrogen pools influencing the observed neutron intensity. At our study site, the calibrated $N_D$ is much higher than the observed average corrected neutron intensity $N_{pi}$ (557 cph). This is probably caused by the influence of the forest vegetation on observed neutron intensities and the calibration parameter of the transfer function and has been similarly described for the standard transfer function by Baatz et al. (2015). As hydrogen stored in air humidity influences the functional relationship between neutron intensities and soil moisture, hydrogen stored in vegetation might have a similar effect. Therefore, a correction or inclusion approach for other above-ground hydrogen pools such as vegetation may yield an even better performance of the UTS and may be investigated in future studies.

Our analyses confirm the improved performance of the UTS compared to the standard transfer function. In order to test whether the improved performance in deriving surface soil moisture translates into a better estimation of soil moisture in deeper layers, we apply the SMAR model using the surface soil moisture time series based on both the revised standard transfer function and the UTS with the MCNP drf parameter set (Fig. 3).



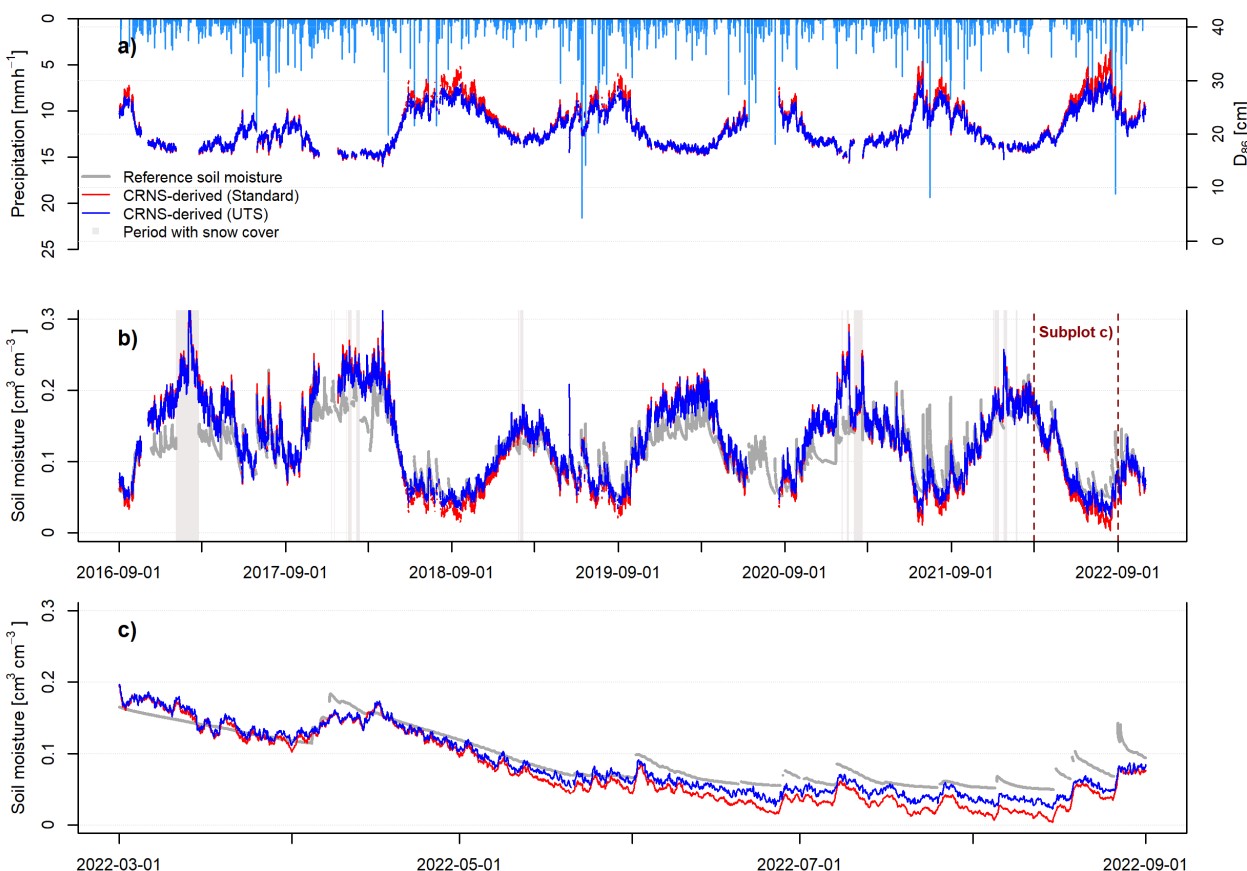

**Figure 3.** Soil moisture estimates with CRNS. (a) estimated time-variable sensitive measurement depth $D_{86}$ of the CRNS-approach and precipitation time series (light blue bars); (b) soil moisture time series derived with the revised standard transfer function and the UTS with parameter set MCNP drf and (c) a period in 2022 illustrating the differences between the two CRNS-derived soil moisture time series.



## 3.2 Depth-extrapolation of CRNS-derived soil moisture time series

### 3.2.1 Original SMAR with calibrated water loss and uncalibrated, modified SMAR

The performance measures and the corresponding values for the depth-extrapolated soil moisture time series based on the calibrated original SMAR (calibrated water loss only) and the uncalibrated modified SMAR (estimated water loss based on eq. (11-14)) are listed in Tab. 3 as well as Tab. 4 and exemplary time series for a second layer depth of 130 cm are shown in Fig. 4 and Fig. 5 for a hourly and daily resolution, respectively. The standard transfer function and the UTS produce similar

results with RMSE values ranging between 0.055 and and 0.015 $cm^3\,cm^{-3}$ for hourly values and between 0.054 and and 0.014 $cm^3\,cm^{-3}$ for daily values over all simulated scenarios. The SMAR model with daily resolution generally results in a higher goodness-of-fit. The correlation coefficients tend to be lower for the scenarios using the uncalibrated, modified SMAR and higher for the original SMAR with calibrated water loss. For KGE, the results are the opposite.

The better performance at a daily time step (irrespective of the depth-extrapolation method) can be attributed to the fact that

it is generally assumed that all water above field capacity infiltrates from the first into the second layer within one time step. While this may be a reasonable assumption on a daily time step for which the SMAR model was designed for (Manfreda et al., 2014), this perquisite is likely to be violated at the hourly time step. Nevertheless, for our study site, the differences in terms of the RMSE are rather small, indicating that the SMAR model may also be used with an hourly resolution.

Following the RMSE threshold of $\leq 0.06$ $cm^3\,cm^{-3}$ which has been used to evaluate the original SMAR performance

(Baldwin et al., 2019; Guo et al., 2023), all simulations with the original and with the modified SMAR and both with an hourly and daily resolution lie below this threshold. This indicates that all SMAR models result in acceptable soil moisture time series for the second soil layer down to 450 cm depth according to RMSE performance. However, taking the dynamic goodness-of-fit parameters KGE and correlation coefficient into account, the performance with regard to the temporal dynamics is not satisfactory. This can also be visually identified from Fig. 4 and Fig. 5 for a second layer depth of 130 cm. The original SMAR

with calibrated constant water loss reaches the wilting point of the second soil layer over large parts of the study period, indicating that the water loss calibrated by minimizing RMSE, results in a high constant water loss to match the reference average water content of the second layer but thereby causing too strong and rapid decreases of soil moisture in dry summer periods. Here, the uncalibrated, modified SMAR model provides more realistic gradual decreases of soil moisture, leading to a better performance when visually assessing the time series. This is also true for the maximum second layer depth of 450 cm

investigated in this study (Fig. A1 and Fig. A2) and illustrates that care should be taken when relying on statistical goodness-of-fit measures and that a visual assessment and interpretation of the simulation results should be undertaken. Nevertheless, it should be noted that the simulated soil moisture time series both for the original and for the modified SMAR do not represent intermediate pulses of increased soil moisture seen in the reference data even during the drier summer period.

For large maximum depths of the second layer such as 450 cm, the original SMAR with calibrated water loss better simulates

the amplitudes of soil moisture in the second layer for a temporal resolution of both hours and days. This indicates that the water loss estimated with the modified SMAR is too low for large depths. The condition of eq. (13), imposing that no evapotranspiration losses occur when water percolates from the first to the second layer, could be one reason. Another reason




could be uncertainties of the relative root fraction that is required to scale the water losses from the the first to the second layer. The use of an exponential model to describe the cumulative root distribution, as done in this study, is highly simplistic and such

models generally remain under debate (e.g. Pierret et al., 2016). Furthermore, too much water percolating from the first into the second soil layer may be compensated through the calibration of the water loss parameter in the original SMAR model, but this cannot be done when the uncalibrated, modified SMAR is applied.

Another major reason for the generally poor performance of both the original and the modified SMAR can be the literature-based soil physical parameters used here in order to apply the SMAR model without calibration against in-situ reference

measurements. In ensemble simulations with the modified SMAR, the soil physical parameters were varied in a range of $\pm$ 10 %. The minimum and maximum RMSE values derived from the 50 hourly and daily ensemble runs are shown in Tab. A1. It can be seen that smaller RMSE values can be achieved with parameter values that are different from the initial ones (Tab. 3). The maximum RMSE for all depths except for 70 cm still meet the RMSE benchmark criterion, indicating a certain robustness of the uncalibrated, modified SMAR model presented in this study if the soil physical parameters can be reasonable well

estimated.

We also tested the impact of the input surface soil moisture time series to both the original SMAR with calibrated water loss and the uncalibrated, modified SMAR. Using either the CRNS-derived soil moisture time series based on the UTS equation or the revised standard equation for the first layer results in visually similar results with similar RMSE values, slightly higher correlation coefficients for the second case, and slightly better KGE values for the first case (Tab. 3, Tab. 4). Overall, in this

study, a better estimated surface soil moisture time series from CRNS does not necessarily translate into a distinct improvement of the depth-extrapolated time series. This may be explained by the considerable overall deficiencies of the SMAR models to represent the soil moisture dynamics at our study site which are larger than the differences between the surface soil moisture time series derived with the different neutron-to-soil moisture transfer functions.

In contrast, improvements of the depth-extrapolated soil moisture times series in the second layer can be seen when the depth

of the top soil layer in the SMAR model is taken to be the median calculated sensitive measurement depth ($D_{86}$, Schrön et al. (2017)) of the CRNS technique. Here, the statistical goodness-of-fit is generally higher compared to using an assumed sensitive depth of 35 cm for the top soil layer. This is the case for both the standard and the modified SMAR model and independent of the transfer function used for the CRNS soil moisture in the top layer (standard or UTS) with hourly and daily resolution. The better matching time series compared to the reference time series is also visible in Fig. 4 and Fig. 5 and is expressed through

the RMSE values in Tab. 3 and Tab. 4. The nature of the SMAR model as a water balance approach implies that the correct estimation of the volume of the upper soil layer and its storage is directly related to the accuracy of the depth-extrapolated time series of the second soil layer. Consequently, an accurate assessment of the sensitive measurement depth of CRNS is also highly important when using CRNS-derived soil moisture time series in e.g. (soil) hydrological model applications.





**Table 3.** Statistical goodness-of-fit between the depth-extrapolated *hourly* soil moisture time series from CRNS surface observations and the average soil moisture time series in the second layer calculated from the available in-situ point-scale soil moisture sensors. The water loss parameter is either a calibrated static value (original SMAR model) or estimated based on the procedure described for the modified SMAR model, see chapter 2.

| Layer 2 depth [cm] | Layer 1 depth [cm] | Transfer function | SMAR | Water loss | $V_2$ [mm h$^{-1}$] | RMSE [cm$^3$ cm$^{-3}$] | Pearson correlation | KGE [-] |
|---|---|---|---|---|---|---|---|---|
| 70 | 35 | Revised standard | Modified | Estimated | - | 0.055 | 0.735 | -1.74 |
| | | | Original | Calibrated | 156 | 0.04 | 0.589 | 0.13 |
| | | UTS MCNP drf | Modified | Estimated | - | 0.053 | 0.733 | -1.64 |
| | | | Original | Calibrated | 132 | 0.041 | 0.577 | 0.14 |
| | 20 | Revised standard | Modified | Estimated | - | 0.041 | 0.803 | -1.54 |
| | | | Original | Calibrated | 64 | 0.035 | 0.661 | 0.24 |
| | | UTS MCNP drf | Modified | Estimated | - | 0.040 | 0.795 | -1.44 |
| | | | Original | Calibrated | 58 | 0.035 | 0.654 | -0.24 |
| 130 | 35 | Revised standard | Modified | Estimated | - | 0.036 | 0.711 | -0.85 |
| | | | Original | Calibrated | 104 | 0.041 | 0.537 | 0.12 |
| | | UTS MCNP drf | Modified | Estimated | - | 0.035 | 0.714 | -0.83 |
| | | | Original | Calibrated | 94 | 0.041 | 0.529 | 0.13 |
| | 20 | Revised standard | Modified | Estimated | - | 0.033 | 0.773 | -0.81 |
| | | | Original | Calibrated | 58 | 0.038 | 0.608 | 0.16 |
| | | UTS MCNP drf | Modified | Estimated | - | 0.033 | 0.768 | -0.79 |
| | | | Original | Calibrated | 52 | 0.038 | 0.604 | 0.17 |
| 200 | 35 | Revised standard | Modified | Estimated | - | 0.033 | 0.698 | -0.82 |
| | | | Original | Calibrated | 113 | 0.036 | 0.492 | 0.15 |
| | | UTS MCNP drf | Modified | Estimated | - | 0.032 | 0.704 | -0.81 |
| | | | Original | Calibrated | 102 | 0.036 | 0.489 | 0.15 |
| | 20 | Revised standard | Modified | Estimated | - | 0.032 | 0.743 | -0.78 |
| | | | Original | Calibrated | 63 | 0.034 | 0.557 | 0.18 |
| | | UTS MCNP drf | Modified | Estimated | - | 0.031 | 0.741 | -0.77 |
| | | | Original | Calibrated | 57 | 0.034 | 0.556 | 0.19 |
| 300 | 35 | Revised standard | Modified | Estimated | - | 0.036 | 0.676 | -1.36 |
| | | | Original | Calibrated | 157 | 0.026 | 0.450 | 0.25 |
| | | UTS MCNP drf | Modified | Estimated | - | 0.035 | 0.681 | -1.33 |
| | | | Original | Calibrated | 142 | 0.026 | 0.449 | 0.25 |
| | 20 | Revised standard | Modified | Estimated | - | 0.034 | 0698 | -1.27 |
| | | | Original | Calibrated | 87 | 0.025 | 0.511 | 0.29 |
| | | UTS MCNP drf | Modified | Estimated | - | 0.033 | 0.694 | -1.24 |
| | | | Original | Calibrated | 79 | 0.025 | 0.512 | 0.29 |
| 450 | 35 | Revised standard | Modified | Estimated | - | 0.044 | 0.545 | -2.02 |
| | | | Original | Calibrated | 300 | 0.015 | 0.333 | 0.3 |
| | | UTS MCNP drf | Modified | Estimated | - | 0.043 | 0.556 | -1.98 |
| | | | Original | Calibrated | 269 | 0.015 | 0.336 | 0.31 |
| | 20 | Revised standard | Modified | Estimated | - | 0.040 | 0.558 | -1.91 |
| | | | Original | Calibrated | 158 | 0.015 | 0.392 | 0.33 |
| | | UTS MCNP drf | Modified | Estimated | - | 0.039 | 0.566 | -1.88 |
| | | | Original | Calibrated | 143 | 0.015 | 0.396 | 0.33 |



**Figure 4.** *Hourly* depth-extrapolated soil moisture time series for a depth of 130 cm using the calibrated standard SMAR model (calibrated water loss $V_2$) with a top layer depth of 35 cm (a), and 20 cm (b) as well as the depth-extrapolated soil moisture time series based on the uncalibrated modified SMAR (estimated water loss) model presented in this study (top layer depth of 35 cm (c) and 20 cm (d)) based on the CRNS-derived surface soil moisture time series from the standard transfer function and the UTS.





**Table 4.** Statistical goodness-of-fit between the depth-extrapolated *daily* surface soil moisture time from CRNS and the average soil moisture time series in the second layer calculated from the available in-situ point-scale soil moisture sensors. The water loss parameter is either as a calibrated static value (original SMAR model) or estimated based on the procedure described in the methods section (modified SMAR model, see chapter 2).

| Layer 2 depth [cm] | Layer 1 depth [cm] | Transfer function | SMAR | Water loss | $V_2$ [mm h$^{-1}$] | RMSE [cm$^3$ cm$^{-3}$] | Pearson correlation | KGE [-] |
|---|---|---|---|---|---|---|---|---|
| 70 | 35 | Revised standard | Modified | Estimated | - | 0.054 | 0.685 | -1.18 |
| | | | Original | Calibrated | 131 | 0.04 | 0.602 | 0.1 |
| | | UTS MCNP drf | Modified | Estimated | - | 0.051 | 0.675 | -0.92 |
| | | | Original | Calibrated | 113 | 0.04 | 0.59 | 0.11 |
| | 20 | Revised standard | Modified | Estimated | - | 0.037 | 0.76 | -1.03 |
| | | | Original | Calibrated | 56 | 0.034 | 0.688 | 0.2 |
| | | UTS MCNP drf | Modified | Estimated | - | 0.035 | 0.752 | -0.78 |
| | | | Original | Calibrated | 51 | 0.034 | 0.678 | 0.22 |
| 130 | 35 | Revised standard | Modified | Estimated | - | 0.034 | 0.626 | -0.38 |
| | | | Original | Calibrated | 87 | 0.039 | 0.575 | 0.08 |
| | | UTS MCNP drf | Modified | Estimated | - | 0.032 | 0.614 | -0.22 |
| | | | Original | Calibrated | 79 | 0.04 | 0.564 | 0.09 |
| | 20 | Revised standard | Modified | Estimated | - | 0.029 | 0.697 | -0.35 |
| | | | Original | Calibrated | 48 | 0.036 | 0.644 | 0.12 |
| | | UTS MCNP drf | Modified | Estimated | - | 0.028 | 0.68 | -0.19 |
| | | | Original | Calibrated | 44 | 0.036 | 0.635 | 0.13 |
| 200 | 35 | Revised standard | Modified | Estimated | - | 0.031 | 0.619 | -0.34 |
| | | | Original | Calibrated | 91 | 0.035 | 0.546 | 0.09 |
| | | UTS MCNP drf | Modified | Estimated | - | 0.03 | 0.608 | -0.19 |
| | | | Original | Calibrated | 83 | 0.035 | 0.538 | 0.1 |
| | 20 | Revised standard | Modified | Estimated | - | 0.027 | 0.685 | -0.3 |
| | | | Original | Calibrated | 51 | 0.032 | 0.611 | 0.13 |
| | | UTS MCNP drf | Modified | Estimated | - | 0.026 | 0.668 | -0.16 |
| | | | Original | Calibrated | 47 | 0.032 | 0.605 | 0.13 |
| 300 | 35 | Revised standard | Modified | Estimated | - | 0.035 | 0.633 | -0.72 |
| | | | Original | Calibrated | 124 | 0.024 | 0.515 | 0.2 |
| | | UTS MCNP drf | Modified | Estimated | - | 0.034 | 0.625 | -0.56 |
| | | | Original | Calibrated | 113 | 0.025 | 0.511 | 0.2 |
| | 20 | Revised standard | Modified | Estimated | - | 0.027 | 0.688 | -0.64 |
| | | | Original | Calibrated | 67 | 0.023 | 0.587 | 0.25 |
| | | UTS MCNP drf | Modified | Estimated | - | 0.027 | 0.673 | -0.48 |
| | | | Original | Calibrated | 62 | 0.023 | 0.583 | 0.25 |
| 450 | 35 | Revised standard | Modified | Estimated | - | 0.043 | 0.551 | -1.18 |
| | | | Original | Calibrated | 237 | 0.014 | 0.392 | 0.37 |
| | | UTS MCNP drf | Modified | Estimated | - | 0.043 | 0.549 | -1.03 |
| | | | Original | Calibrated | 215 | 0.014 | 0.392 | 0.37 |
| | 20 | Revised standard | Modified | Estimated | - | 0.032 | 0.594 | -1.1 |
| | | | Original | Calibrated | 122 | 0.014 | 0.47 | 0.39 |
| | | UTS MCNP drf | Modified | Estimated | - | 0.032 | 0.581 | -0.95 |
| | | | Original | Calibrated | 112 | 0.014 | 0.47 | 0.39 |





**Figure 5.** *Daily* depth-extrapolated soil moisture time series for a depth of 130 cm using the calibrated standard SMAR model (calibrated water loss $V_2$) with a top layer depth of 35 cm (a), and 20 cm (b) as well as the depth-extrapolated soil moisture time series based on the uncalibrated modified SMAR (estimated water loss) model presented in this study (top layer depth of 35 cm (c) and 20 cm (d)) based on the CRNS-derived surface soil moisture time series from the standard transfer function and the UTS.



### 3.2.2 Full calibration of the original and modified SMAR

To further assess the performance of the original and modified SMAR at the study site, we performed a full (all parameter) calibration of the two SMAR models with 10,000 random combinations of the soil physical parameters. The initially assigned soil physical parameters were altered in the range of $\pm\, 20\,\%$ to assign values in a physically acceptable range for the sandy soils at the study site. Additionally to the soil physical parameters, the bulk water loss $V_2$ in the original SMAR was calibrated with random values in the range from 1 to 500 mm. For the modified SMAR model, the relative root fraction in the first layer

$R_1$ was varied in the range of $\pm\, 20\,\%$ instead.

The results of the full calibration can be found in Tab. 5 and Tab. 6 and exemplary time series for a second layer depth of 130 cm are shown in Fig. 6 and Fig. 7 for a hourly and daily resolution, respectively. The results for the maximum depth of 450 cm can be found in the appendix. As expected, minimizing the RMSE in the calibration period 2017 leads to a decrease of the RMSE for the entire study period compared to the uncalibrated modified SMAR or when calibrating the water loss term

in the original SMAR only. This is the case for both the hourly and daily time step, with generally better performance for the latter in terms of RMSE and KGE. Using a first layer depth of 20 cm instead of 35 cm leads to better soil moisture dynamics in the deeper layers. This is in line with results presented in the previous chapters when comparing the uncalibrated, modified SMAR and the original SMAR with calibrated water loss.

Following the statistical goodness-of-fit parameters in Tab. 5 and Tab. 6, the modified SMAR performs worse than the

calibrated original SMAR in different depths after all parameters have been calibrated. This may be attributed to using a single-objective optimization for minimizing the RMSE, only. Furthermore, in the original SMAR, the bulk water loss from the second layer was optimized while for the modified SMAR, only the cumulative root fraction in the first layer was adjusted. This leads to more restricted conditions for the modified SMAR model. For example, calibrating the estimated complete water loss in the latter eq. 11 based on a calibration factor could lead to improved the results of the fully calibrated, modified SMAR

and more close to those derived for the fully calibrated original SMAR. Nevertheless, the generally higher process restrictions due to the defined estimation of $ET_2$ and $P_2$ in eq. 13–14 of the modified SMAR remain.

In summary, the full calibration of the original and modified SMAR model with soil model parameters in physically reasonable value range show similar characteristics to those scenarios described in the previous chapter where literature-based values where assigned for soil physical parameters and only the bulk water loss $V_2$ was calibrated. Although the overall vi-

sual performance improved and a higher statistical goodness-of-fit can be achieved when all environmental model parameters are calibrated, the original and modified SMAR model tested in this study do not show satisfying results with respect to the temporal dynamics of the soil moisture time series of the second layer. Many intermediate rainfall events are not captured and thus, the reference soil moisture time series show a more dynamic behavior than those simulated by the original and modified SMAR.

Calibration experiments revealed that assigning values in a non-physical parameter range, e.g., negative values, for soil physical model parameters could lead to an improved performance of SMAR. When allowing a non-physical value range for the model parameters $sc_1$, $sc_2$ and $sw_2$, the visual and statistical performance of both, the original and modified SMAR improve





dramatically with the exception of the depth of 70 cm. Again exemplary shown for a maximum depth of 130 and 450 cm depth
and both temporal resolutions, Fig. A5 – A8 as well as Tab. A2 – A3 illustrate the improved performance of both SMAR
models. The poor and even worse performance in the depth of 70 cm compared to the full calibration with physically reasonable
values could be related to a non-sufficient value range for the calibrated parameters when values in a non-physically based value
range are assigned. Even better results may also be derived for other depths with a different value range or by also calibrating
the remaining parameters $n_1$, $n_2$, $V_2$ and $R_1$ in a non-physically reasonable value range. These results show that satisfactory
results with the original and the modified SMAR can be obtained at our study site at the expense of physical realism of the
model, and only if in-situ soil moisture measurements in the depth of interest are available for calibration.

### 3.2.3   General discussion

The evaluation of the original SMAR model against in-situ observations in previous studies showed a range of RMSE values
and correlation coefficients (e.g., Manfreda et al., 2014; Faridani et al., 2017), indicating that the performance of the SMAR
model varies between study sites.

The particular water flow dynamics at our study site located in a mixed forest with sandy soils may explain the overall un-
satisfactory representation of soil moisture dynamics of the SMAR model when model parameters are assigned in a physically
reasonable range. Preferential flow in macropores including bypass flow along roots (e.g., Nimmo, 2021) can result in highly
conductive forest soils with infiltrating water being quickly transported from the surface to deeper layers. For example, Chan-
dler et al. (2018) and Alaoui et al. (2011) found that forest soils can have higher saturated hydraulic conductivity compared
to other land cover types and combined with differing preferential flow processes this may lead to increased infiltration and
percolation into lower layers of forest soils (e.g., Alaoui et al., 2011). Complex preferential flow and infiltration processes
are unlikely to be properly captured by the SMAR as it allows water movement only for soil moisture conditions above field
capacity. A more complex root distribution than the exponential one assumed in this study and related temporally varying tran-
spiration water losses from different depths adds for complexity that is not captured by the SMAR model. Maysonnave et al.
(2022), for instance, found that root water uptake in forests can vary with time and depth depending on the water availability in
different layers. These features can neither be reproduced by the original nor modified SMAR and makes forest sites generally
challenging, in particular for simplified models. However, this may be partly compensated for when model-specific effective
parameters are used. In this case, calibration against in-situ reference soil moisture information is required and the parameters
lose their physical meaning and interpretability but may account for the particular soil hydraulic processes of the study site.

In addition to the simplicity of the model, the field-scale approach of this study adds further difficulties when evaluating
the simulated soil moisture time series against point-scale soil moisture observations. The reason is their high spatio-temporal
variability, especially in forests caused by e.g. heterogeneous evapotranspiration, interception, (e.g., Schume et al., 2003) and
root distribution patterns (e.g., Jost et al., 2012). Even more, the decreasing number of reference in-situ soil moisture sensors
with increasing soil depth may lead to a lower representativeness of the reference soil moisture time series at larger depths,
lowering comparability to the model results. Nevertheless, with point sensors down to 450 cm, this study allows for exploring
the potential of SMAR for larger depths than usually feasible. Even when depths down to 450 cm are considered, the original



and modified SMAR meet the benchmark RMSE of $\leq 0.06$ cm$^3$ cm$^{-3}$ in scenarios with literature-based and with calibrated model parameters. This underlines the usefulness of SMAR to derive a first estimate of soil moisture in a second, deeper soil layer.

The largest limitation of the present study for evaluating the standard and the introduced modified SMAR models is its application to a single observation site. A comparison with other simple depth-extrapolation approaches including the soil water index (e.g., Wagner et al., 1999; Albergel et al., 2008), empirical approaches such as regression models (e.g., Zhang et al., 2017) and cumulative distribution function matching (e.g., Gao et al., 2018) as well as other versions of the SMAR model (e.g., Faridani et al., 2017) would allow for an improved evaluation of the presented modification of the SMAR model

and should be assessed in future studies at sites with a broader range of climatic conditions, vegetation covers and soils.





**Table 5.** Statistical goodness-of-fit between the depth-extrapolated *hourly* surface soil moisture time series from CRNS and the average soil moisture time series in the second layer calculated from the available in-situ point-scale soil moisture sensors with the fully calibrated SMAR in a physically acceptable parameter range. The calibrated model parameters and goodness-of-fit indicators for the original and modified SMAR model are shown.

| Layer 2 depth [cm] | Layer 1 depth [cm] | Transfer function | SMAR | $n_1$ [-] | $n_2$ [-] | $sc_1$ [cm³ cm⁻³] | $sc_2$ [cm³ cm⁻³] | $sw_2$ [cm³ cm⁻³] | $V_2$ [mm h⁻¹] | $R_1$ [-] | RMSE [cm³ cm⁻³] | Pearson correlation | KGE [-] |
|---|---|---|---|---|---|---|---|---|---|---|---|---|---|
| 70 | 35 | Revised standard | Modified | 0.56 | 0.34 | 0.19 | 0.10 | 0.04 | - | 0.77 | 0.030 | 0.584 | -0.286 |
| | | | Original | 0.50 | 0.36 | 0.13 | 0.11 | 0.05 | 377 | - | 0.029 | 0.709 | -0.029 |
| | | UTS MCNP drf | Modified | 0.59 | 0.33 | 0.19 | 0.10 | 0.04 | - | 0.78 | 0.027 | 0.587 | -0.148 |
| | | | Original | 0.50 | 0.36 | 0.13 | 0.11 | 0.05 | 377 | - | 0.028 | 0.707 | 0.072 |
| | 20 | Revised standard | Modified | 0.68 | 0.41 | 0.19 | 0.1 | 0.04 | - | 0.54 | 0.028 | 0.748 | -0.062 |
| | | | Original | 0.64 | 0.45 | 0.12 | 0.13 | 0.05 | 123 | - | 0.024 | 0.754 | 0.263 |
| | | UTS MCNP drf | Modified | 0.67 | 0.42 | 0.18 | 0.10 | 0.05 | - | 0.54 | 0.025 | 0.736 | 0.041 |
| | | | Original | 0.64 | 0.45 | 0.12 | 0.13 | 0.05 | 123 | - | 0.024 | 0.753 | 0.342 |
| 130 | 35 | Revised standard | Modified | 0.55 | 0.35 | 0.19 | 0.10 | 0.04 | - | 0.77 | 0.031 | 0.565 | -0.014 |
| | | | Original | 0.51 | 0.41 | 0.13 | 0.13 | 0.05 | 226 | - | 0.028 | 0.678 | 0.364 |
| | | UTS MCNP drf | Modified | 0.56 | 0.42 | 0.19 | 0.10 | 0.05 | - | 0.78 | 0.024 | 0.557 | 0.169 |
| | | | Original | 0.57 | 0.36 | 0.13 | 0.10 | 0.05 | 136 | - | 0.028 | 0.681 | 0.093 |
| | 20 | Revised standard | Modified | 0.63 | 0.45 | 0.18 | 0.10 | 0.05 | - | 0.53 | 0.026 | 0.692 | 0.175 |
| | | | Original | 0.64 | 0.45 | 0.12 | 0.13 | 0.05 | 123 | - | 0.026 | 0.724 | 0.372 |
| | | UTS MCNP drf | Modified | 0.51 | 0.41 | 0.17 | 0.10 | 0.05 | - | 0.53 | 0.025 | 0.707 | 0.145 |
| | | | Original | 0.64 | 0.45 | 0.12 | 0.13 | 0.05 | 123 | - | 0.026 | 0.727 | 0.438 |
| 200 | 35 | Revised standard | Modified | 0.54 | 0.33 | 0.19 | 0.1 | 0.04 | - | 0.74 | 0.026 | 0.576 | 0.113 |
| | | | Original | 0.51 | 0.41 | 0.13 | 0.13 | 0.05 | 226 | - | 0.025 | 0.634 | 0.378 |
| | | UTS MCNP drf | Modified | 0.60 | 0.40 | 0.19 | 0.10 | 0.04 | - | 0.77 | 0.023 | 0.574 | 0.164 |
| | | | Original | 0.51 | 0.41 | 0.13 | 0.13 | 0.05 | 226 | - | 0.024 | 0.642 | 0.436 |
| | 20 | Revised standard | Modified | 0.63 | 0.45 | 0.18 | 0.10 | 0.05 | - | 0.53 | 0.023 | 0.661 | 0.175 |
| | | | Original | 0.64 | 0.45 | 0.12 | 0.13 | 0.05 | 123 | - | 0.023 | 0.676 | 0.301 |
| | | UTS MCNP drf | Modified | 0.51 | 0.41 | 0.17 | 0.10 | 0.05 | - | 0.52 | 0.023 | 0.682 | 0.135 |
| | | | Original | 0.64 | 0.45 | 0.12 | 0.13 | 0.05 | 123 | - | 0.023 | 0.684 | 0.375 |
| 300 | 35 | Revised standard | Modified | 0.52 | 0.31 | 0.19 | 0.10 | 0.05 | - | 0.61 | 0.019 | 0.575 | -0.047 |
| | | | Original | 0.50 | 0.36 | 0.13 | 0.11 | 0.05 | 377 | - | 0.017 | 0.576 | 0.533 |
| | | UTS MCNP drf | Modified | 0.52 | 0.31 | 0.19 | 0.10 | 0.05 | - | 0.61 | 0.019 | 0.561 | 0.062 |
| | | | Original | 0.51 | 0.45 | 0.13 | 0.14 | 0.05 | 370 | - | 0.016 | 0.585 | 0.523 |
| | 20 | Revised standard | Modified | 0.63 | 0.31 | 0.19 | 0.10 | 0.05 | - | 0.43 | 0.020 | 0.591 | 0.059 |
| | | | Original | 0.76 | 0.42 | 0.13 | 0.14 | 0.05 | 230 | - | 0.016 | 0.617 | 0.576 |
| | | UTS MCNP drf | Modified | 0.59 | 0.31 | 0.18 | 0.10 | 0.05 | - | 0.48 | 0.019 | 0.592 | 0.108 |
| | | | Original | 0.68 | 0.36 | 0.12 | 0.10 | 0.05 | 136 | - | 0.015 | 0.629 | 0.512 |
| 450 | 35 | Revised standard | Modified | 0.51 | 0.32 | 0.19 | 0.10 | 0.04 | - | 0.59 | 0.023 | 0.425 | -0.851 |
| | | | Original | 0.52 | 0.31 | 0.17 | 0.11 | 0.05 | 482 | - | 0.011 | 0.287 | 0.075 |
| | | UTS MCNP drf | Modified | 0.52 | 0.31 | 0.19 | 0.10 | 0.05 | - | 0.61 | 0.018 | 0.451 | -0.234 |
| | | | Original | 0.63 | 0.31 | 0.16 | 0.12 | 0.05 | 485 | - | 0.010 | 0.328 | 0.12 |
| | 20 | Revised standard | Modified | 0.63 | 0.31 | 0.19 | 0.14 | 0.05 | - | 0.43 | 0.018 | 0.466 | -0.268 |
| | | | Original | 0.66 | 0.34 | 0.13 | 0.14 | 0.05 | 484 | - | 0.009 | 0.484 | 0.272 |
| | | UTS MCNP drf | Modified | 0.63 | 0.31 | 0.19 | 0.10 | 0.05 | - | 0.43 | 0.017 | 0.461 | -0.145 |
| | | | Original | 0.66 | 0.34 | 0.13 | 0.14 | 0.05 | 484 | - | 0.009 | 0.496 | 0.261 |







**Figure 6.** *Hourly* depth-extrapolated soil moisture time series for a depth of 130 cm using the standard SMAR model with a top layer depth of 35 cm (a), and 20 cm (b) as well as the depth-extrapolated soil moisture time series based on the modified SMAR model presented in this study (top layer depth of 35 cm (c) and 20 cm (d)) based on the CRNS-derived surface soil moisture time series from the standard transfer function and the UTS. The soil physical parameters $n_1$, $n_2$, $sc_1$, $sc_2$, $sw_2$ and $R_1$ were optimized by reducing the RMSE against reference soil moisture values in the year 2017. For the original SMAR model, the water loss term $V_2$ was calibrated instead of $R_1$.





**Table 6.** Statistical goodness-of-fit between the depth-extrapolated *daily* surface soil moisture time from CRNS and the average soil moisture time series in the second layer calculated from the available in-situ point-scale soil moisture sensors with the fully calibrated SMAR in a physically acceptable parameter range. The calibrated model parameters and goodness-of-fit indicators for the original and modified SMAR model are shown.

| Layer 2 depth [cm] | Layer 1 depth [cm] | Transfer function | SMAR | $n_1$ [-] | $n_2$ [-] | $sc_1$ [cm³ cm⁻³] | $sc_2$ [cm³ cm⁻³] | $sw_2$ [cm³ cm⁻³] | $V_2$ [mm h⁻¹] | $R_1$ [-] | RMSE [cm³ cm⁻³] | Pearson correlation | KGE [-] |
|---|---|---|---|---|---|---|---|---|---|---|---|---|---|
| 70 | 35 | Revised standard | Modified | 0.54 | 0.33 | 0.19 | 0.10 | 0.04 | - | 0.74 | 0.024 | 0.518 | 0.119 |
| | | | Original | 0.50 | 0.36 | 0.13 | 0.11 | 0.05 | 377 | - | 0.029 | 0.710 | 0.011 |
| | | UTS MCNP drf | Modified | 0.47 | 0.35 | 0.19 | 0.10 | 0.03 | - | 0.78 | 0.022 | 0.490 | 0.252 |
| | | | Original | 0.50 | 0.36 | 0.13 | 0.11 | 0.05 | 377 | - | 0.028 | 0.707 | 0.111 |
| | 20 | Revised standard | Modified | 0.72 | 0.40 | 0.18 | 0.1 | 0.04 | - | 0.54 | 0.021 | 0.681 | 0.228 |
| | | | Original | 0.64 | 0.45 | 0.12 | 0.13 | 0.05 | 123 | - | 0.024 | 0.766 | 0.316 |
| | | UTS MCNP drf | Modified | 0.59 | 0.45 | 0.18 | 0.10 | 0.05 | - | 0.55 | 0.020 | 0.655 | 0.307 |
| | | | Original | 0.64 | 0.45 | 0.12 | 0.13 | 0.05 | 123 | - | 0.023 | 0.763 | 0.389 |
| 130 | 35 | Revised standard | Modified | 0.55 | 0.35 | 0.19 | 0.10 | 0.04 | - | 0.77 | 0.021 | 0.441 | 0.321 |
| | | | Original | 0.57 | 0.36 | 0.13 | 0.10 | 0.05 | 136 | - | 0.028 | 0.694 | 0.086 |
| | | UTS MCNP drf | Modified | 0.58 | 0.44 | 0.19 | 0.10 | 0.04 | - | 0.78 | 0.019 | 0.390 | 0.358 |
| | | | Original | 0.57 | 0.36 | 0.13 | 0.10 | 0.05 | 136 | - | 0.028 | 0.694 | 0.176 |
| | 20 | Revised standard | Modified | 0.59 | 0.45 | 0.18 | 0.10 | 0.05 | - | 0.55 | 0.021 | 0.585 | 0.351 |
| | | | Original | 0.63 | 0.34 | 0.12 | 0.12 | 0.05 | 63 | - | 0.026 | 0.741 | 0.031 |
| | | UTS MCNP drf | Modified | 0.55 | 0.33 | 0.15 | 0.10 | 0.04 | - | 0.54 | 0.022 | 0.689 | 0.165 |
| | | | Original | 0.63 | 0.34 | 0.12 | 0.12 | 0.05 | 163 | - | 0.025 | 0.741 | 0.121 |
| 200 | 35 | Revised standard | Modified | 0.54 | 0.33 | 0.19 | 0.10 | 0.04 | - | 0.74 | 0.019 | 0.425 | 0.334 |
| | | | Original | 0.51 | 0.41 | 0.13 | 0.13 | 0.05 | 226 | - | 0.024 | 0.657 | 0.437 |
| | | UTS MCNP drf | Modified | 0.45 | 0.44 | 0.19 | 0.10 | 0.04 | - | 0.75 | 0.018 | 0.321 | 0.304 |
| | | | Original | 0.57 | 0.36 | 0.13 | 0.10 | 0.05 | 136 | - | 0.024 | 0.665 | 0.162 |
| | 20 | Revised standard | Modified | 0.59 | 0.45 | 0.18 | 0.10 | 0.05 | - | 0.55 | 0.019 | 0.554 | 0.352 |
| | | | Original | 0.64 | 0.45 | 0.12 | 0.13 | 0.05 | 123 | - | 0.022 | 0.705 | 0.390 |
| | | UTS MCNP drf | Modified | 0.59 | 0.45 | 0.18 | 0.10 | 0.05 | - | 0.55 | 0.019 | 0.503 | 0.383 |
| | | | Original | 0.64 | 0.45 | 0.12 | 0.13 | 0.05 | 123 | - | 0.022 | 0.708 | 0.448 |
| 300 | 35 | Revised standard | Modified | 0.56 | 0.34 | 0.19 | 0.10 | 0.04 | - | 0.77 | 0.017 | 0.377 | 0.141 |
| | | | Original | 0.53 | 0.44 | 0.13 | 0.11 | 0.05 | 323 | - | 0.016 | 0.605 | 0.504 |
| | | UTS MCNP drf | Modified | 0.52 | 0.41 | 0.19 | 0.10 | 0.04 | - | 0.75 | 0.017 | 0.310 | 0.223 |
| | | | Original | 0.53 | 0.44 | 0.13 | 0.11 | 0.05 | 323 | - | 0.016 | 0.610 | 0.538 |
| | 20 | Revised standard | Modified | 0.68 | 0.41 | 0.19 | 0.10 | 0.04 | - | 0.54 | 0.018 | 0.446 | 0.171 |
| | | | Original | 0.68 | 0.36 | 0.12 | 0.10 | 0.05 | 136 | - | 0.015 | 0.665 | 0.536 |
| | | UTS MCNP drf | Modified | 0.52 | 0.41 | 0.18 | 0.11 | 0.05 | - | 0.54 | 0.017 | 0.376 | 0.178 |
| | | | Original | 0.68 | 0.36 | 0.13 | 0.10 | 0.05 | 136 | - | 0.015 | 0.669 | 0.574 |
| 450 | 35 | Revised standard | Modified | 0.52 | 0.31 | 0.19 | 0.10 | 0.05 | - | 0.61 | 0.016 | 0.362 | 0.05 |
| | | | Original | 0.63 | 0.31 | 0.16 | 0.12 | 0.05 | 485 | - | 0.01 | 0.365 | 0.156 |
| | | UTS MCNP drf | Modified | 0.52 | 0.31 | 0.19 | 0.10 | 0.05 | - | 0.61 | 0.016 | 0.291 | 0.028 |
| | | | Original | 0.62 | 0.34 | 0.16 | 0.10 | 0.05 | 492 | - | 0.01 | 0.363 | 0.170 |
| | 20 | Revised standard | Modified | 0.63 | 0.31 | 0.19 | 0.10 | 0.05 | - | 0.43 | 0.013 | 0.416 | 0.189 |
| | | | Original | 0.66 | 0.34 | 0.13 | 0.14 | 0.05 | 484 | - | 0.009 | 0.517 | 0.283 |
| | | UTS MCNP drf | Modified | 0.63 | 0.31 | 0.19 | 0.10 | 0.05 | - | 0.43 | 0.013 | 0.384 | 0.217 |
| | | | Original | 0.66 | 0.34 | 0.13 | 0.14 | 0.05 | 484 | - | 0.009 | 0.527 | 0.270 |







**Figure 7.** *Daily* depth-extrapolated soil moisture time series for a depth of 130 cm using the standard SMAR model with a top layer depth of 35 cm (a), and 20 cm (b) as well as the depth-extrapolated soil moisture time series based on the modified SMAR model presented in this study (top layer depth of 35 cm (c) and 20 cm (d)) based on the CRNS-derived surface soil moisture time series from the standard transfer function and the UTS. The soil physical parameters $n_1$, $n_2$, $sc_1$, $sc_2$, $sw_2$ and $R_1$ were optimized by reducing the RMSE against reference soil moisture values in the year 2017. For the original SMAR model, the water loss term $V_2$ was calibrated instead of $R_1$.





## 4 Conclusions

In the present study we investigated the feasibility of depth-extrapolating surface soil moisture time series derived from CRNS to deeper soil layers without additional in-situ soil moisture information for calibration. We furthermore evaluated the Universal Transport Solution (UTS) for the estimation of field scale soil moisture from CRNS neutron counts.

Being among the first who evaluate the UTS as a new transfer function to estimate field-scale surface soil moisture information from CRNS, we confirm its improved performance compared to the standard approach. The UTS accounts for the interdependence of soil moisture and air humidity on the observed neutron intensity, being most important for dry soil conditions. Although applied at a forested site with rather dry soils but with large amounts of above-ground hydrogen stored in the local biomass and influencing the neutron signal, CRNS-derived soil moisture estimates can be improved compared to using established transfer functions. Thus, our results suggest hat the UTS should be used for an improved estimation of surface soil moisture in future CRNS research and applications.

We modified SMAR for estimating soil moisture times series in a second, deeper layer in a way that it can be applied without calibration against in-situ sensors and with soil physical properties and the cumulative root fraction as a vegetation parameter only. Our analyses show that for a benchmark RMSE of $\leq 0.06\,\mathrm{cm}^3\,\mathrm{cm}^{-3}$, the uncalibrated modified SMAR can compete with the original SMAR model down to a maximum depth of the second soil layer of $450\,\mathrm{cm}$ when the same soil physical properties are assigned and only the water loss term is calibrated. A certain robustness of the uncalibrated, modified SMAR in terms of the RMSE was shown by sensitivity runs of the model. However, major temporal dynamics of the reference in-situ soil moisture in the second soil layer are neither captured by original nor by the modified SMAR. This is likely linked to the location of the study site in a mixed forest site with sandy soils, accompanied with preferential flow and root water uptake processes that are difficult to simulate, especially with rather simple modeling approaches. Only the use of SMAR with calibrated effective albeit non-physical parameters partly accommodates to the specific soil hydraulic processes at the study site, showing an improved simulation of soil moisture dynamics in a second soil layer. Under these circumstances, deeper soil moisture time series may be more satisfactorily simulated even with simple modeling approaches such as SMAR.

Although our study suggests that improved surface soil moisture estimates from CRNS do not translate to distinctly improved soil moisture estimates in greater depths, a more accurate estimation of the representative measurement depth of CRNS leads to better results of the SMAR model. This indicates that an accurate estimation of the representative measurement depth of CRNS is especially important when using CRNS data as input for hydrological models.

Given the overall performance of the SMAR model at our single study site, further research and testing of the presented modified version of the SMAR model with and without calibration at sites with varying climatic conditions, vegetation cover and soil properties is necessary and encouraged for future studies. Despite the overall unsatisfactory performance of the SMAR model with respect to accurately capturing soil moisture dynamics at our study site, meeting the defined RMSE benchmark, the simple modification of the SMAR algorithm may serve as a valuable first estimate of soil moisture from a second, deeper soil layer, when in-situ reference soil moisture information for calibration are not available and the soil physical parameters can be reasonably well estimated.



In CRNS research, this modified SMAR approach opens up potential for roving CRNS, i.e., by mounting CRNS instruments on cars (e.g., Schrön et al., 2018a) or trains (e.g., Schrön et al., 2021; Altdorff et al., 2023) moving beyond the field-scale of stationary CRNS applications, thereby providing valuable information for landscape water balancing or hydrological catchment models on larger scales. Moreover, the modified SMAR approach introduced in this study is not limited to CRNS applications. It may also be used in estimating root-zone soil moisture in greater depths from satellite derived surface soil moisture in which
the original SMAR already proved useful (e.g., Baldwin et al., 2017, 2019; Gheybi et al., 2019).

*Data availability.* All data sets are available from the authors upon request.

**Appendix A**

**Table A1.** Minimum and maximum RMSE values between the depth-extrapolated soil moisture time from CRNS using the modified SMAR model of the 50 ensemble runs and the reference soil moisture time series in the second layer calculated from the available in-situ point-scale soil moisture sensors for the simulations with hourly and daily resolution.

| | | | Hourly resolution | | Daily resolution | |
|---|---|---|---|---|---|---|
| **Transfer function** | **Layer 1 depth [cm]** | **Layer 2 depth [cm]** | **$RMSE_{min}$ [cm$^3$ cm$^{-3}$]** | **$RMSE_{max}$ [cm$^3$ cm$^{-3}$]** | **$RMSE_{min}$ [cm$^3$ cm$^{-3}$]** | **$RMSE_{max}$ [cm$^3$ cm$^{-3}$]** |
| | | 70 | 0.040 | 0.073 | 0.037 | 0.072 |
| | | 130 | 0.029 | 0.049 | 0.025 | 0.049 |
| | 35 | 200 | 0.027 | 0.044 | 0.023 | 0.045 |
| | | 300 | 0.026 | 0.047 | 0.025 | 0.047 |
| | | 450 | 0.034 | 0.057 | 0.032 | 0.058 |
| Revised standard | | | | | | |
| | | 70 | 0.032 | 0.052 | 0.027 | 0.049 |
| | | 130 | 0.027 | 0.042 | 0.024 | 0.038 |
| | 20 | 200 | 0.026 | 0.04 | 0.023 | 0.034 |
| | | 300 | 0.025 | 0.043 | 0.02 | 0.035 |
| | | 450 | 0.031 | 0.052 | 0.023 | 0.043 |
| | | 70 | 0.037 | 0.070 | 0.035 | 0.070 |
| | | 130 | 0.028 | 0.048 | 0.024 | 0.049 |
| | 35 | 200 | 0.027 | 0.043 | 0.022 | 0.045 |
| | | 300 | 0.024 | 0.047 | 0.024 | 0.047 |
| | | 450 | 0.032 | 0.057 | 0.032 | 0.059 |
| UTS MCNP drf | | | | | | |
| | | 70 | 0.031 | 0.051 | 0.026 | 0.049 |
| | | 130 | 0.027 | 0.041 | 0.023 | 0.037 |
| | 20 | 200 | 0.026 | 0.04 | 0.022 | 0.033 |
| | | 300 | 0.024 | 0.043 | 0.019 | 0.035 |
| | | 450 | 0.03 | 0.051 | 0.023 | 0.044 |







**Figure A1.** *Hourly* depth-extrapolated soil moisture time series for a depth of 450 cm using the calibrated standard SMAR model (calibrated water loss $V_2$) with a top layer depth of 35 cm (a), and 20 cm (b) as well as the depth-extrapolated soil moisture time series based on the uncalibrated modified SMAR (estimated water loss) model presented in this study (top layer depth of 35 cm (c) and 20 cm (d)) based on the CRNS-derived surface soil moisture time series from the standard transfer function and the UTS.



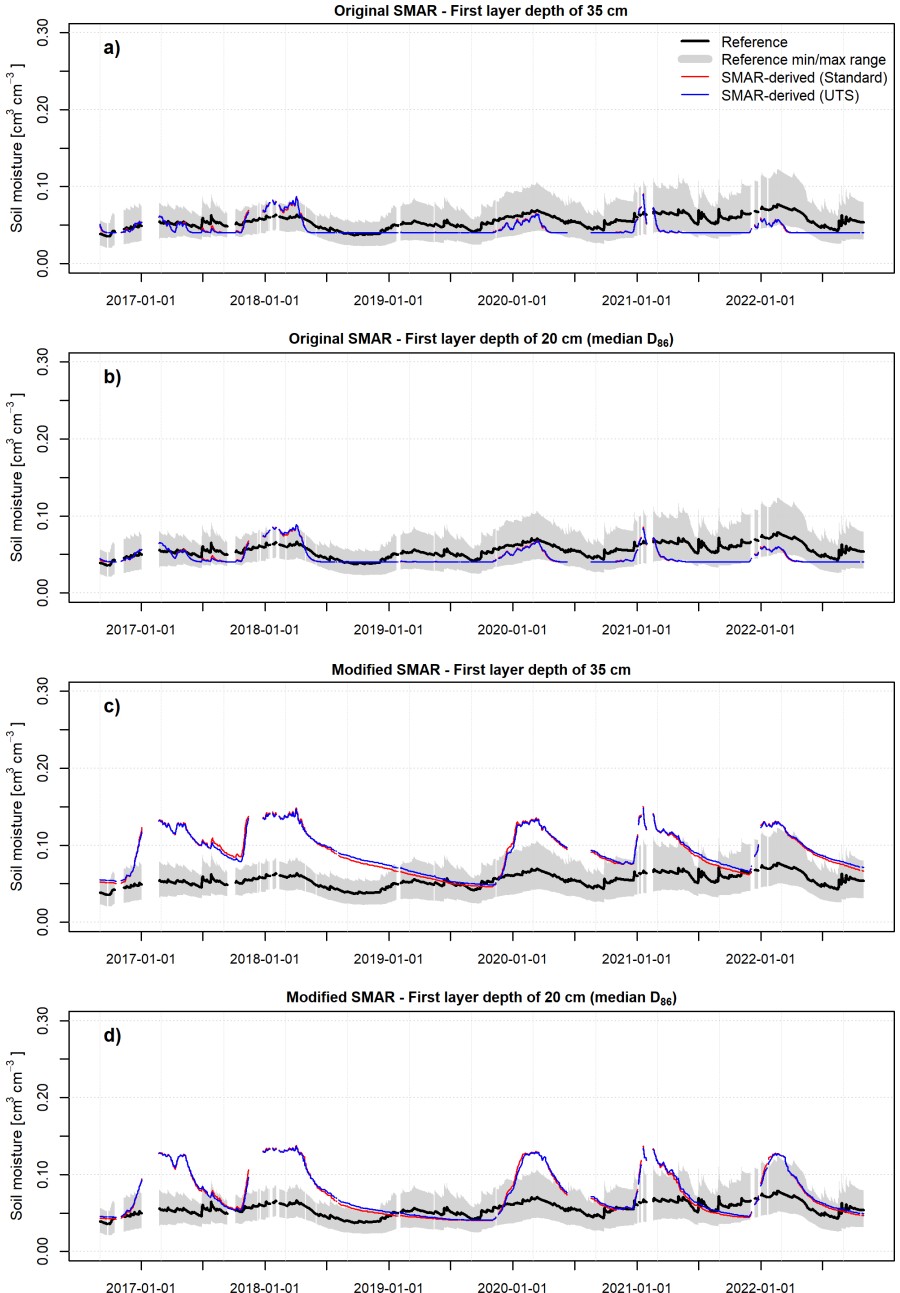

**Figure A2.** *Daily* depth-extrapolated soil moisture time series for a depth of 450 cm using the calibrated standard SMAR model (calibrated water loss $V_2$) with a top layer depth of 35 cm (a), and 20 cm (b) as well as the depth-extrapolated soil moisture time series based on the uncalibrated modified SMAR (estimated water loss) model presented in this study (top layer depth of 35 cm (c) and 20 cm (d)) based on the CRNS-derived surface soil moisture time series from the standard transfer function and the UTS.




**Figure A3.** *Hourly* depth-extrapolated soil moisture time series for a depth of 450 cm using the standard SMAR model with a top layer depth of 35 cm (a), and 20 cm (b) as well as the depth-extrapolated soil moisture time series based on the modified SMAR model presented in this study (top layer depth of 35 cm (c) and 20 cm (d)) based on the CRNS-derived surface soil moisture time series from the standard transfer function and the UTS. The soil physical parameters $n_1$, $n_2$, $sc_1$, $sc_2$, $sw_2$ and $R_1$ were optimized by reducing the RMSE against reference soil moisture values in the year 2017. For the original SMAR model, the water loss term $V_2$ was calibrated instead of $R_1$.





**Figure A4.** *Daily* depth-extrapolated soil moisture time series for a depth of 450 cm using the standard SMAR model with a top layer depth of 35 cm (a), and 20 cm (b) as well as the depth-extrapolated soil moisture time series based on the modified SMAR model presented in this study (top layer depth of 35 cm (c) and 20 cm (d)) based on the CRNS-derived surface soil moisture time series from the standard transfer function and the UTS. The soil physical parameters $n_1$, $n_2$, $sc_1$, $sc_2$, $sw_2$ and $R_1$ were optimized by reducing the RMSE against reference soil moisture values in the year 2017. For the original SMAR model, the water loss term $V_2$ was calibrated instead of $R_1$.





**Table A2.** Statistical goodness-of-fit between the depth-extrapolated *hourly* surface soil moisture time series from CRNS and the average soil moisture time series in the second layer calculated from the available in-situ point-scale soil moisture sensors with the fully calibrated SMAR and effective parameters in a non-physically based value range. The calibrated model parameters and goodness-of-fit indicators for the original and modified SMAR model are shown.

| Layer 2 depth [cm] | Layer 1 depth [cm] | Transfer function | SMAR | $n_1$ [-] | $n_2$ [-] | $sc_1$ [cm$^3$ cm$^{-3}$] | $sc_2$ [cm$^3$ cm$^{-3}$] | $sw_2$ [cm$^3$ cm$^{-3}$] | $V_2$ [mm h$^{-1}$] | $R_1$ [-] | RMSE [cm$^3$ cm$^{-3}$] | Pearson correlation | KGE [-] |
|---|---|---|---|---|---|---|---|---|---|---|---|---|---|
| 70 | 35 | Revised standard | Modified | 0.44 | 0.33 | -0.25 | -0.96 | -0.45 | - | 0.77 | 0.057 | 0.847 | -2.014 |
| | | | Original | 0.55 | 0.35 | -0.03 | -0.25 | -0.10 | 393 | - | 0.054 | 0.846 | -1.948 |
| | | UTS MCNP drf | Modified | 0.44 | 0.37 | 0.04 | -0.88 | -0.02 | - | 0.65 | 0.045 | 0.836 | -1.488 |
| | | | Original | 0.55 | 0.35 | -0.03 | -0.25 | -0.1 | 393 | - | 0.047 | 0.836 | -1.552 |
| | 20 | Revised standard | Modified | 0.73 | 0.36 | -0.72 | -0.88 | -0.38 | - | 0.51 | 0.015 | 0.849 | 0.458 |
| | | | Original | 0.59 | 0.46 | 0.02 | -0.3 | 0.03 | 452 | - | 0.016 | 0.849 | 0.423 |
| | | UTS MCNP drf | Modified | 0.73 | 0.36 | -0.72 | -0.88 | -0.38 | - | 0.51 | 0.013 | 0.841 | 0.602 |
| | | | Original | 0.64 | 0.31 | 0 | -0.07 | 0.03 | 353 | - | 0.012 | 0.841 | 0.627 |
| 130 | 35 | Revised standard | Modified | 0.42 | 0.38 | -0.96 | -0.87 | -0.54 | - | 0.73 | 0.016 | 0.839 | 0.447 |
| | | | Original | 0.54 | 0.34 | 0.01 | -0.22 | 0.01 | 358 | - | 0.025 | 0.838 | 0.007 |
| | | UTS MCNP drf | Modified | 0.42 | 0.38 | -0.96 | -0.87 | -0.54 | - | 0.73 | 0.014 | 0.829 | 0.558 |
| | | | Original | 0.54 | 0.34 | 0.01 | -0.22 | 0.01 | 358 | - | 0.022 | 0.829 | 0.199 |
| | 20 | Revised standard | Modified | 0.66 | 0.38 | -0.53 | -0.09 | -0.75 | - | 0.45 | 0.008 | 0.855 | 0.790 |
| | | | Original | 0.64 | 0.37 | -0.23 | -0.48 | -0.01 | 497 | - | 0.009 | 0.855 | 0.839 |
| | | UTS MCNP drf | Modified | 0.66 | 0.38 | -0.53 | -0.09 | -0.75 | - | 0.45 | 0.009 | 0.847 | 0.717 |
| | | | Original | 0.64 | 0.37 | -0.23 | -0.48 | -0.01 | 497 | - | 0.009 | 0.846 | 0.781 |
| 200 | 35 | Revised standard | Modified | 0.45 | 0.31 | -0.79 | -0.58 | -0.19 | - | 0.68 | 0.011 | 0.794 | 0.782 |
| | | | Original | 0.53 | 0.31 | 0 | -0.07 | 0.03 | 353 | - | 0.015 | 0.794 | 0.366 |
| | | UTS MCNP drf | Modified | 0.45 | 0.31 | -0.79 | -0.58 | -0.19 | - | 0.68 | 0.010 | 0.788 | 0.750 |
| | | | Original | 0.53 | 0.31 | 0 | -0.07 | 0.03 | 353 | - | 0.013 | 0.512 | 0.787 |
| | 20 | Revised standard | Modified | 0.71 | 0.38 | -0.68 | -0.33 | -0.07 | - | 0.51 | 0.009 | 0.811 | 0.563 |
| | | | Original | 0.54 | 0.32 | -0.90 | -0.75 | -0.22 | 479 | - | 0.014 | 0.811 | 0.739 |
| | | UTS MCNP drf | Modified | 0.71 | 0.38 | -0.68 | -0.33 | -0.07 | - | 0.51 | 0.009 | 0.805 | 0.504 |
| | | | Original | 0.75 | 0.37 | -0.96 | -0.53 | -0.27 | 498 | - | 0.012 | 0.805 | 0.749 |
| 300 | 35 | Revised standard | Modified | 0.49 | 0.36 | -0.93 | -0.60 | -0.12 | - | 0.71 | 0.009 | 0.733 | 0.658 |
| | | | Original | 0.54 | 0.34 | 0.01 | -0.22 | 0.01 | 358 | - | 0.022 | 0.734 | -0.042 |
| | | UTS MCNP drf | Modified | 0.49 | 0.36 | -0.93 | -0.60 | -0.12 | - | 0.71 | 0.009 | 0.728 | 0.602 |
| | | | Original | 0.54 | 0.34 | 0.01 | -0.22 | 0.01 | 358 | - | 0.019 | 0.729 | 0.147 |
| | 20 | Revised standard | Modified | 0.61 | 0.33 | -0.79 | -0.02 | -0.85 | - | 0.49 | 0.009 | 0.751 | 0.393 |
| | | | Original | 0.70 | 0.35 | -0.09 | -0.73 | 0.02 | 374 | - | 0.009 | 0.751 | 0.739 |
| | | UTS MCNP drf | Modified | 0.61 | 0.33 | -0.79 | -0.02 | -0.85 | - | 0.49 | 0.009 | 0.746 | 0.350 |
| | | | Original | 0.70 | 0.35 | -0.09 | -0.73 | 0.02 | 374 | - | 0.009 | 0.747 | 0.738 |
| 450 | 35 | Revised standard | Modified | 0.55 | 0.32 | -0.24 | -0.44 | 0.01 | - | 0.65 | 0.010 | 0.593 | 0.448 |
| | | | Original | 0.54 | 0.37 | 0.06 | -0.42 | 0.02 | 448 | - | 0.019 | 0.577 | 0.045 |
| | | UTS MCNP drf | Modified | 0.55 | 0.32 | -0.24 | -0.44 | 0.01 | - | 0.65 | 0.010 | 0.592 | 0.404 |
| | | | Original | 0.54 | 0.37 | 0.06 | -0.42 | 0.02 | 448 | - | 0.018 | 0.586 | 0.139 |
| | 20 | Revised standard | Modified | 0.54 | 0.44 | -0.27 | -0.70 | 0.03 | - | 0.46 | 0.009 | 0.619 | 0.232 |
| | | | Original | 0.64 | 0.37 | -0.23 | -0.48 | -0.01 | 497 | - | 0.009 | 0.618 | 0.609 |
| | | UTS MCNP drf | Modified | 0.54 | 0.44 | -0.27 | -0.70 | 0.03 | - | 0.46 | 0.009 | 0.618 | 0.202 |
| | | | Original | 0.64 | 0.37 | -0.23 | -0.48 | -0.01 | 497 | - | 0.008 | 0.619 | 0.616 |



**Table A3.** Statistical goodness-of-fit between the depth-extrapolated *daily* surface soil moisture time from CRNS and the average soil moisture time series in the second layer calculated from the available in-situ point-scale soil moisture sensors with the fully calibrated SMAR and effective parameters in a non-physically based value range. The calibrated model parameters and goodness-of-fit indicators for the original and modified SMAR model are shown.

| Layer 2 depth [cm] | Layer 1 depth [cm] | Transfer function | SMAR | $n_1$ [-] | $n_2$ [-] | $sc_1$ [cm³ cm⁻³] | $sc_2$ [cm³ cm⁻³] | $sw_2$ [cm³ cm⁻³] | $V_2$ [mm h⁻¹] | $R_1$ [-] | RMSE [cm³ cm⁻³] | Pearson correlation | KGE [-] |
|---|---|---|---|---|---|---|---|---|---|---|---|---|---|
| 70 | 35 | Revised standard | Modified | 0.42 | 0.43 | 0.11 | -0.82 | 0.04 | - | 0.61 | 0.033 | 0.764 | -0.555 |
| | | | Original | 0.45 | 0.37 | 0.01 | 0.34 | -0.05 | 488 | - | 0.052 | 0.854 | -1.806 |
| | | UTS MCNP drf | Modified | 0.48 | 0.31 | 0.09 | -0.76 | 0.04 | - | 0.64 | 0.032 | 0.804 | -0.743 |
| | | | Original | 0.45 | 0.37 | 0.01 | -0.34 | -0.05 | 488 | - | 0.046 | 0.844 | -1.42 |
| | 20 | Revised standard | Modified | 0.74 | 0.33 | 0 | -0.77 | 0.03 | - | 0.54 | 0.013 | 0.856 | 0.521 |
| | | | Original | 0.74 | 0.34 | -0.06 | -0.69 | 0 | 452 | - | 0.015 | 0.856 | 0.482 |
| | | UTS MCNP drf | Modified | 0.74 | 0.33 | 0 | -0.77 | 0.03 | - | 0.54 | 0.012 | 0.848 | 0.658 |
| | | | Original | 0.68 | 0.33 | -0.02 | -0.03 | 0.02 | 391 | - | 0.012 | 0.848 | 0.619 |
| 130 | 35 | Revised standard | Modified | 0.61 | 0.33 | 0 | -0.77 | 0.03 | - | 0.78 | 0.016 | 0.843 | 0.536 |
| | | | Original | 0.48 | 0.32 | 0.04 | -0.86 | 0.04 | 477 | - | 0.013 | 0.840 | 0.378 |
| | | UTS MCNP drf | Modified | 0.60 | 0.31 | -0.43 | -0.80 | -0.16 | - | 0.63 | 0.013 | 0.835 | 0.641 |
| | | | Original | 0.48 | 0.32 | 0.04 | -0.86 | 0.04 | 477 | - | 0.016 | 0.835 | 0.502 |
| | 20 | Revised standard | Modified | 0.55 | 0.39 | -0.68 | -0.14 | -0.70 | - | 0.46 | 0.008 | 0.862 | 0.798 |
| | | | Original | 0.64 | 0.37 | -0.23 | -0.48 | -0.01 | 497 | - | 0.009 | 0.846 | 0.846 |
| | | UTS MCNP drf | Modified | 0.73 | 0.43 | -0.29 | -0.03 | -0.69 | - | 0.46 | 0.008 | 0.854 | 0.769 |
| | | | Original | 0.65 | 0.38 | -0.04 | -0.64 | 0.04 | 423 | - | 0.008 | 0.854 | 0.808 |
| 200 | 35 | Revised standard | Modified | 0.47 | 0.31 | -0.33 | -0.71 | -0.03 | - | 0.73 | 0.01 | 0.798 | 0.775 |
| | | | Original | 0.61 | 0.34 | -0.06 | -0.69 | 0 | 452 | - | 0.018 | 0.804 | 0.383 |
| | | UTS MCNP drf | Modified | 0.47 | 0.31 | -0.33 | -0.71 | -0.03 | - | 0.73 | 0.010 | 0.792 | 0.727 |
| | | | Original | 0.61 | 0.34 | -0.06 | -0.69 | 0 | 452 | - | 0.016 | 0.797 | 0.527 |
| | 20 | Revised standard | Modified | 0.72 | 0.42 | -0.93 | -0.19 | -0.29 | - | 0.43 | 0.009 | 0.820 | 0.617 |
| | | | Original | 0.73 | 0.33 | -0.19 | -0.13 | 0.01 | 421 | - | 0.009 | 0.822 | 0.769 |
| | | UTS MCNP drf | Modified | 0.59 | 0.45 | -0.39 | -0.03 | -0.39 | - | 0.39 | 0.009 | 0.814 | 0.608 |
| | | | Original | 0.61 | 0.31 | -0.08 | -0.23 | 0.04 | 409 | - | 0.009 | 0.814 | 0.644 |
| 300 | 35 | Revised standard | Modified | 0.55 | 0.40 | -0.54 | -1 | -0.03 | - | 0.56 | 0.008 | 0.741 | 0.626 |
| | | | Original | 0.48 | 0.32 | 0.04 | -0.86 | 0.04 | 477 | - | 0.012 | 0.747 | 0.502 |
| | | UTS MCNP drf | Modified | 0.55 | 0.40 | -0.54 | -1 | -0.03 | - | 0.56 | 0.008 | 0.736 | 0.567 |
| | | | Original | 0.48 | 0.32 | 0.04 | -0.86 | 0.04 | 477 | - | 0.011 | 0.744 | 0.597 |
| | 20 | Revised standard | Modified | 0.57 | 0.46 | -0.28 | 0.02 | -0.57 | - | 0.40 | 0.008 | 0.764 | 0.504 |
| | | | Original | 0.67 | 0.33 | -0.39 | -0.49 | -0.04 | 442 | - | 0.009 | 0.771 | 0.760 |
| | | UTS MCNP drf | Modified | 0.66 | 0.39 | 0.06 | 0.06 | -0.39 | - | 0.50 | 0.024 | 0.666 | -0.429 |
| | | | Original | 0.67 | 0.33 | -0.39 | -0.49 | -0.04 | 442 | - | 0.008 | 0.766 | 0.747 |
| 450 | 35 | Revised standard | Modified | 0.55 | 0.34 | -0.20 | -0.91 | 0.02 | - | 0.74 | 0.009 | 0.604 | 0.405 |
| | | | Original | 0.42 | 0.35 | 0 | -0.17 | 0.01 | 487 | - | 0.016 | 0.619 | 0.067 |
| | | UTS MCNP drf | Modified | 0.44 | 0.37 | -0.82 | -0.83 | -0.04 | - | 0.54 | 0.008 | 0.604 | 0.387 |
| | | | Original | 0.42 | 0.35 | 0 | -0.17 | 0.01 | 487 | - | 0.015 | 0.617 | 0.23 |
| | 20 | Revised standard | Modified | 0.71 | 0.39 | -1 | -0.04 | -0.47 | - | 0.41 | 0.008 | 0.637 | 0.339 |
| | | | Original | 0.64 | 0.31 | -0.33 | -0.13 | -0.03 | 456 | - | 0.011 | 0.649 | 0.618 |
| | | UTS MCNP drf | Modified | 0.54 | 0.34 | -0.65 | 0 | -0.66 | - | 0.41 | 0.008 | 0.633 | 0.260 |
| | | | Original | 0.64 | 0.31 | -0.33 | -0.13 | -0.03 | 456 | - | 0.010 | 0.647 | 0.627 |



**Figure A5.** *Hourly* depth-extrapolated soil moisture time series for a depth of $130\,\mathrm{cm}$ using the standard SMAR model with a top layer depth of $35\,\mathrm{cm}$ (a), and $20\,\mathrm{cm}$ (b) as well as the depth-extrapolated soil moisture time series based on the modified SMAR model presented in this study (top layer depth of $35\,\mathrm{cm}$ (c) and $20\,\mathrm{cm}$ (d)) based on the CRNS-derived surface soil moisture time series from the standard transfer function and the UTS. The soil physical parameters $n_1$, $n_2$, $sc_1$, $sc_2$, $sw_2$ and $R_1$ were optimized by reducing the RMSE against reference soil moisture values in the year 2017. Here, the parameters $sc_1$, $sc_2$ and $sw_2$ were calibrated as effective parameters in a non-physically based value range. For the original SMAR model, the water loss term $V_2$ was calibrated instead of $R_1$.



**Figure A6.** *Daily* depth-extrapolated soil moisture time series for a depth of 130 cm using the standard SMAR model with a top layer depth of 35 cm (a), and 20 cm (b) as well as the depth-extrapolated soil moisture time series based on the modified SMAR model presented in this study (top layer depth of 35 cm (c) and 20 cm (d)) based on the CRNS-derived surface soil moisture time series from the standard transfer function and the UTS. The soil physical parameters $n_1$, $n_2$, $sc_1$, $sc_2$, $sw_2$ and $R_1$ were optimized by reducing the RMSE against reference soil moisture values in the year 2017. Here, the parameters $sc_1$, $sc_2$ and $sw_2$ were calibrated as effective parameters in a non-physically based value range. For the original SMAR model, the water loss term $V_2$ was calibrated instead of $R_1$.





**Figure A7.** *Hourly* depth-extrapolated soil moisture time series for a depth of 450 cm using the standard SMAR model with a top layer depth of 35 cm (a), and 20 cm (b) as well as the depth-extrapolated soil moisture time series based on the modified SMAR model presented in this study (top layer depth of 35 cm (c) and 20 cm (d)) based on the CRNS-derived surface soil moisture time series from the standard transfer function and the UTS. The soil physical parameters $n_1$, $n_2$, $sc_1$, $sc_2$, $sw_2$ and $R_1$ were optimized by reducing the RMSE against reference soil moisture values in the year 2017. Here, the parameters $sc_1$, $sc_2$ and $sw_2$ were calibrated as effective parameters in a non-physically based value range. For the original SMAR model, the water loss term $V_2$ was calibrated instead of $R_1$.



**Figure A8.** *Daily* depth-extrapolated soil moisture time series for a depth of 450 cm using the standard SMAR model with a top layer depth of 35 cm (a), and 20 cm (b) as well as the depth-extrapolated soil moisture time series based on the modified SMAR model presented in this study (top layer depth of 35 cm (c) and 20 cm (d)) based on the CRNS-derived surface soil moisture time series from the standard transfer function and the UTS. The soil physical parameters $n_1$, $n_2$, $sc_1$, $sc_2$, $sw_2$ and $R_1$ were optimized by reducing the RMSE against reference soil moisture values in the year 2017. Here, the parameters $sc_1$, $sc_2$ and $sw_2$ were calibrated as effective parameters in a non-physically based value range. For the original SMAR model, the water loss term $V_2$ was calibrated instead of $R_1$.



*Author contributions.* DR further developed the original ideas of TB and AG for this study, performed the data analysis and wrote the manuscript. TB and AG designed the soil moisture monitoring network and contributed to the writing of the manuscript.

*Competing interests.* The authors declare no competing interests.

*Acknowledgements.* This study was conducted as part of the research unit Cosmic Sense funded by the German Research Foundation (Deutsche Forschungsgemeinschaft, DFG-FOR2694, project no. 357874777). We gratefully acknowledge the technical support of Markus Morgner, Jörg Wummel and Stephan Schröder who maintain the observation sites in TERENO-NE funded by the Helmholtz Association. In addition, we would like to thank Paul Voit for his assistance in data acquisition, field and laboratory work. Further, we would like to thank the
Müritz National Park for the continuing support and collaboration. Lastly, we acknowledge the NMDB database (www.nmdb.eu) founded under the European Union's FP7 programme (contract no. 213007), and the PIs of individual neutron monitors for providing data.



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
