# Peer review of "Depth-extrapolation of field-scale soil moisture time series derived with cosmic-ray neutron sensing using the SMAR model"

_EGUsphere, 2024_

## Referee Comment (RC1)

Review of the manuscript egusphere-2024-170, Depth-extrapolation of field-scale soil moisture time series derived with cosmic-ray neutron sensing using the SMAR model by Rasche et al.

**Summary**

The Authors explore the possibility to estimate deep soil moisture (SM) from cosmic-ray neutron sensing (CRNS) observations based on the SMAR model. A good data-set from a point-scale soil moisture network (SMN) is used as a benchmark. Several alternatives are explored by combing different CRNS-based SM and the SMAR models settings. The results show that the CRNS- SMAR approach is not able to reproduce depth-extrapolated SM well, but it is argued that it is in line with previous studies (i.e. RMSE < 0.06 m3 m-3).

The manuscript is well structured but not always clear. Some additional information should be reported. Several analyses have been performed but I did not recognize a clear design to address the research question. Many factors are in fact combined and it is not clear if the performances depend on the CRNS, on the unrepresentativeness of the SMN, or on the SMAR model. So, in my opinion readers are stuck on the outcome of the study and how to proceed. Overall, I believe that the Authors should put major effort into improving the manuscript for possible publication. Below I provide more details to clarify my arguments.

**Major concerns**

**[1] About the use of point scale soil moisture network (SMN)**

The soil moisture network (SMN) represents a valuable data-sets (L131-132). Still, it is explained that could not represent CRNS footprint well (L348). Considering in addition that the deeper soil moisture network (> 40 cm) is based on a lower number of point-scale sensors, it could be questionable its value as benchmark for testing the performance of the depth-extrapolated CRNS-SM. For this reason, it cannot be concluded in my opinion if the difficulties to estimated deeper SM are based on CRNS settings, SMAR settings or an unrepresentativeness of the SMN. To circumvent this issue, I strongly recommend the Authors first to test SMAR settings only to the SMN, i.e., using first layer of SM measured by the SMN to estimate the deeper SM. Then, to apply the best settings of the SMAR model to CRNS. This exercise could be performed to each single soil moisture profile and any combinations. The results should provide a benchmark for testing the depth-extrapolated SM based on CRNS. Please note that you have already quantified some differences between CRNS and SMN. Thus, it should be expected that the depth extrapolated CRNS-SM cannot perform better than that. Finally, it is not clear to me if the depth extrapolated CRNS is compared with the weighted SMN or to the arithmetic average SMN. It should be the former to be consistent with the signal but with the disadvantage to have again weighted soil

moisture. This issue seems to not be resolvable and could limit some applications. Discussion should integrate this aspect.

**[2] About the use of SMAR**

First, I found misleading the arguments to support the use of SMAR. The Authors argue several times that "most approaches require a site-specific calibration using depth-profiles of in-situ soil moisture data, which are often not available." (L6-7). For this reason, you use SMAR. However, this approach also requires calibration. So your arguments do not support your decision. Similarly, it is stated (L9) that "SMAR is usually also calibrated to sensor data, but could be applied without calibration if all its parameters were known" and L90. "This method does not require calibration if the environmental parameters are known." Well this is valid for any models and not only for SMAR. The problem is that it is often the case that we do not know the parameters, and especially at the scale of model application when soil is under investigation. The results of this study also confirm this statement. So your arguments do not support your decision to use SMAR as many other approaches have similar settings. Overall, I'm fine with the decision to test SMAR (L86). But I'm against the argument that this is physically based and it does not require adjustment in comparison to other approaches. The results of the present study also confirm my doubt. Comparison to other approaches, e.g., the exponential filter (Franz et al., 2020; Wagner et al., 1999) could strengthen this study.

Finally, the Authors tested several SMAR settings. Despite the overall exercises are remarkable, I'm wondering why testing approaches that already sound not appropriate. I.e.,I found the assumptions behind the original SMAR model very arguable. E.g., a constant flux V2 (P and ET) is already debated when applied for estimating monthly groundwater recharge and it should be considered unrealistic over a long time series (years) for hourly to daily resolutions. The results show how V2 moves from, e.g., 50 to 300 mm h-1 in Table 3. These results are not consistent and support my doubt of using standard approach. So I'm not surprised if the Authors tried to modify the approach and I would remove the scenario with the constant V2. Similarly, the CRNS effective depth seems to be easily estimated (L279). So why testing a guess of 35 cm? I would remove this scenario. Overfitting a model calibrating all the parameters also seems to me a crash test and it does not seem to me a good approach for testing if a model is reliable. Several parameters are not in a physically consistent range and what was argued to be a physical approach has been destroyed. I would also remove this scenario.

**Specific comments**

L3-5. deeper than what? I would rephrase otherwise it is not clear to what you refer when saying "deeper", e.g., moving the sentence "Many of these applications require information on soil water dynamics in deeper soil layers." After " Cosmic-ray neutron sensing (CRNS) allows for non-invasive monitoring of field-scale soil moisture across several hectares around the instrument but only for the first few tens of centimeters of the soil."

L8. Personally, I'm not too much on the philosophical discussion about process-based or empirical approaches (e.g., (Hrachowitz and Clark, 2017)). But to call a bucket model with several empirical assumptions a physical-based approach could be highly criticized. Please consider rewording.

L36. Despite, I agree that establishing an extensive point sensor network requires a lot of effort, the international soil moisture network ISMN (Dorigo et al., 2021) provides a good example that it is worth citing.

L80 and L82. To my knowledge, (Wagner et al., 1999) proposed the use of an exponential filter approach to estimate soil water content at deeper soil layers based on surface soil moisture. Instead the term SWI refers to a quantity of water content between 0-100%. Thus, I would not call the extrapolation approach "soil water index" but rather "exponential filter approach".

L128. How field capacity and wilting point have been estimated is missing and should be added.

L134. The use of the manufacture's calibration function could be another source of errors that should be discussed in the result section.

L154. The fact that you use a 25h moving average could be a reason to not be able to capture fast soil moisture changes? This could be discussed in the result section.

Eq.5. I guess h is air humidity. Please clarify.

L179. Please clarify how look-up-table approach works. What are also the assumptions/approximations in using this approach?

L181. So how Nd has been actually calculated? Is it related to N0? Please clarify.

L187. What are the differences between URANOS type and MCNP type equation?

L211. Are you sure you have to move back and forward from volumetric soil moisture and relative saturation? Could you not try directly expanding eq. 7 directly for volumetric soil moisture?

L248. I guess the estimation would be very sensitive to the length of the time series. Did you use the entire time series for estimating ET? Please clarify.

L490. Well, the discovery that root water uptake varies with time and depth depending on the water availability goes probably back to the introduction of irrigation practices in the history. I guess Maysonnave et al.,2022 showed more than that. If the Authors remove the scenario with V2 constant, this comment could also be probably removed. Otherwise please rephrase.

The Authors have stated that this study is the first who evaluates the UTS (L515). If this is the case I think is worth clarifying this from the beginning and provide some additional information to better understand how it works and is implemented.

**References**

Dorigo, W., Himmelbauer, I., Aberer, D., Schremmer, L., Petrakovic, I., Zappa, L., Preimesberger, W., Xaver, A., Annor, F., Ardö, J., Baldocchi, D., Bitelli, M., Blöschl, G., Bogena, H., Brocca, L., Calvet, J.-C., Camarero, J.J., Capello, G., Choi, M., Cosh, M.C., van de Giesen, N., Hajdu, I., Ikonen, J., Jensen, K.H., Kanniah, K.D., de Kat, I., Kirchengast, G., Kumar Rai, P., Kyrouac, J., Larson, K., Liu, S., Loew, A., Moghaddam, M., Martínez Fernández, J., Mattar Bader, C., Morbidelli, R., Musial, J.P., Osenga, E., Palecki, M.A., Pellarin, T., Petropoulos, G.P., Pfeil, I., Powers, J., Robock, A., Rüdiger, C., Rummel, U., Strobel, M., Su, Z., Sullivan, R., Tagesson, T., Varlagin, A., Vreugdenhil, M., Walker, J., Wen, J., Wenger, F., Wigneron, J.P., Woods, M., Yang, K., Zeng, Y., Zhang, X., Zreda, M., Dietrich, S., Gruber, A., van Oevelen, P., Wagner, W., Scipal, K., Drusch, M., Sabia, R., 2021. The International Soil Moisture Network: serving Earth system science for over a decade. Hydrology and Earth System Sciences 25, 5749–5804. https://doi.org/10.5194/hess-25-5749-2021

Franz, T.E., Wahbi, A., Zhang, J., Vreugdenhil, M., Heng, L., Dercon, G., Strauss, P., Brocca, L., Wagner, W., 2020. Practical Data Products From Cosmic-Ray Neutron Sensing for Hydrological Applications. Front. Water 2, 9. https://doi.org/10.3389/frwa.2020.00009

Hrachowitz, M., Clark, M.P., 2017. HESS Opinions: The complementary merits of competing modelling philosophies in hydrology. Hydrol. Earth Syst. Sci. 21, 3953–3973. https://doi.org/10.5194/hess-21-3953-2017

Wagner, W., Lemoine, G., Rott, H., 1999. A Method for Estimating Soil Moisture from ERS Scatterometer and Soil Data. Remote Sensing of Environment 70, 191–207. https://doi.org/10.1016/S0034-4257(99)00036-X

---

## Author Comment (AC2)

**Reviewer #2:**

We thank reviewer #2 for taking the time to review our manuscript and the helpful comments which will improve the quality of the manuscript. In the following section we will reply to all comments of reviewer #2 by numbering reviewer comments starting with R2-1 (i.e. reviewer 2, comment 1) and our response with A2-1 (i.e. author response to R2-1).

**R2-1:** MAJOR: A large number of simulations have been carried out. It is difficult to make a summary of the results obtained and to clearly understand the more important scientific understanding that has been gained by carrying out the study. For example, when looking at the figures, it seems that no configuration gives satisfactory results, especially in terms of soil moisture dynamics. A RMSE lower than 0.06 cm^3/cm^3 is not very representative and depends on the dynamic range of the soil moisture data. I would strongly suggest reducing the number of simulations and focusing on more relevant results.

**A2-1:** We are thankful about this comment and this leads into the same direction compared to what reviewer #1 mentioned in his/her comments. As also suggested by reviewer #1, we will remove the scenarios with hourly data, a first layer depth of 35 cm and all scenarios in which we calibrated all soil physical parameters. This will help to streamline our analyses and make the main points in our manuscript clearer.

**R2-1:** MAJOR: Why are the results not as good as expected? Does it depend on the SMAR model, or on the CRNS data, or on the soil moisture benchmark? This should be clarified. For example, a simulation with surface and deeper layers from the Soil Moisture Benchmark data can give insight into the performance of SMAR. Some additional analysis in this direction should be done.

**A2-2:** This is again a very similar comment to what has been mentioned by reviewer #1. We are glad to see that the improvements which need to be made on this manuscript are clear.
In addition to removing a large part of the scenarios as mentioned in A2-1, we will perform the remaining scenarios of SMAR also based on the in-situ point-scale soil moisture sensors, only. This means that we will use the surface soil moisture time series of the in-situ soil moisture sensors and apply all SMAR scenarios for two selected soil

moisture sensor profiles. Furthermore, we will then perform the analyses based on depth averages from the entire soil moisture sensor network.

This will give us a better, site-specific benchmark to evaluate the original and modified version of the SMAR model by testing it on in-situ profile scale and in-situ network scale before testing the influence of a CRNS-derived soil moisture time series as the input. This way, we may be also able to get more clues on the performance of the original SMAR model and our modified version.

**R2-3:** MODERATE: Other methods have been proposed to extrapolate soil moisture data from the surface to deeper layers. The exponential filter is the most commonly used approach. I would suggest a comparison with such an approach, again to provide additional information on the performance of the SMAR model.

**A2-3:** The reviewer is right. The exponential filter method (also referred to as the Soil Moisture Index) is a famous and often used approach to derive soil moisture time series in deeper layers from a surface soil moisture time series. We initially decided not to add this approach to our manuscript as its only bulk fitting parameter T is not directly linked to a single specific physical quantity and is difficult to predict for individual observation sites (e.g. Zhang et al. 2017, Wang et al. 2017). Hence, it always requires calibration against reference soil moisture information in the depth of interest. However, finding an approach which can be used without reference information in the depth of interest was the initial objective of this study.

Nevertheless, as also pointed out by reviewer #1, the exponential filter is a standard approach and comparing the original SMAR with calibrated water loss parameter $V_2$ and the completely uncalibrated modified SMAR with the exponential filter with calibrated T parameter can provide some additional insights into the performance of either SMAR version at our study site.

Following the response to the comments of reviewer #1, we will repeat all analyses also with the exponential filter and add the results to a modified version of our manuscript.

**Additional references:**

Wang, T., Franz, T. E., You, J., Shulski, M. D., and Ray, C.: Evaluating controls of soil properties and climatic conditions on the use of an exponential filter for converting near surface to root zone soil moisture contents, Journal of Hydrology, 548, 683-696, http://dx.doi.org/10.1016/j.jhydrol.2017.03.055, 2017.

---

## Author Response (AR2)

**Revision of the manuscript "egusphere-2024-170" submitted to SOIL**

This document contains a point-by-point reply to the comments of all reviewers, corresponding author responses and adjustments made in the original manuscript.

The major comments of the two anonymous reviewers concerned the number of analyses conducted, the structure of the manuscript as well as the application of depth-extrapolation approaches and appropriate references for comparison. The reviewers suggested to enhance the structure of this paper by removing certain depth-extrapolation scenarios. Furthermore, recommendations were made to also apply the exponential filter method in addition to the SMAR model and the modified SMAR model and to use it as a reference when evaluating the performance of both SMAR models.

We followed the comments of the reviewers by rephrasing large sections of the manuscript. We re-structured our data analyses and removed a large fraction of the original analyses making the results easier to follow. We added the exponential filter as an additional depth-extrapolation approach for comparison with our analyses and added a short chapter to the methods section. We modified large parts of chapter 2.3.2 (now 2.3.3) and completely replaced chapter 3.2. The introduction and conclusion
chapter were adjusted accordingly.
The changes made in the revised manuscript also include the introduction of several new figures and a reduction of tables which helps to better follow the results and their interpretation and discussion.

The individual reviewer comments are denoted with R1-1 (reviewer 1, comment 1) and A1-1 (author response to reviewer 1,
comment 1). When changes are made to phrases in the original manuscript, we state the original and adjusted sentence/paragraph or provide the phrase added.

Please not that given the large number of changes made in the revised manuscript including the replacement of entire chapters. The revised chapters 2.3.2, 2.3.3 and 3.2 including all new figures are added to the end of this point-to-point reply. For all
further changes (including reduced result tables in the appendix) which are neither addressed in the individual reviewer comments below nor at the end of this point-to-point reply, please see the track-changes-document.

We thank reviewer #1 for taking the time to review our manuscript. Following the suggestions of reviewer #1, we agree that our manuscript needed major revisions before it can be published. We are certain that the reviewer's comments helped to streamline our manuscript, to make it clearer and more relevant for the reader of SOIL.

**R1-1:** The soil moisture network (SMN) represents a valuable data-sets (L131-132). Still, it is explained that could not represent CRNS footprint well (L348). Considering in addition that the deeper soil moisture network (> 40 cm) is based on a lower number of point-scale sensors, it could be questionable its value as benchmark for testing the performance of the depth-extrapolated CRNS-SM. For this reason, it cannot be concluded in my opinion if the difficulties to estimated deeper SM are based on CRNS settings, SMAR settings or an unrepresentativeness of the SMN. To circumvent this issue, I strongly recommend the Authors first to test SMAR settings only to the SMN, i.e., using first layer of SM measured by the SMN to estimate the deeper SM. Then, to apply the best settings of the SMAR model to CRNS. This exercise could be performed to each single soil moisture profile and any combinations. The results should provide a benchmark for testing the depth-extrapolated SM based on CRNS. Please note that you have already quantified some differences between CRNS and SMN.

Thus, it should be expected that the depth extrapolated CRNS-SM cannot perform better than that. Finally, it is not clear to me if the depth extrapolated CRNS is compared with the weighted SMN or to the arithmetic average SMN. It should be the former to be consistent with the signal but with the disadvantage to have again weighted soil moisture. This issue seems to not be resolvable and could limit some applications. Discussion should integrate this aspect.

**A1-1:** We fully agree that a decreasing number is reference in-situ point-scale soil moisture sensors is likely to reduce the representativeness of the SMN in these depths. This indeed adds to the uncertainties coming with the surface soil moisture time series derived from CRNS as well as the SMAR model and the defined soil physical parameters used in SMAR. Different potential reasons for the poor performance are already given in the manuscript but they are difficult to disentangle. We agree that further analyses are necessary and agree on testing the original and modified version of SMAR on the SMN only prior using it with CRNS-derived surface soil moisture time series. For this reason, we conducted the following additional analyses and added them to the revised manuscript:

1) We performed a comparison of the original and modified SMAR using the soil moisture in 0-20 cm depth for two individual soil moisture profiles which comprise sensors in all depths (i.e. 10, 20, 30, 50, 70, 130, 200, 300 and 450 cm). The SMAR- derived second layer soil moisture time series was then compared to the soil moisture information in the respective depths of the two sensor profiles.

2) We performed a comparison of the original and modified SMAR using the average soil moisture in 0-20 cm depth from the entire SMN. The SMAR-derived second layer soil moisture time series was then compared to the average soil moisture information in the respective depths of the SMN.

These additional analyses were performed on daily time steps and with the literature-based soil physical parameters shown in Tab. 1. This means that for the original SMAR, only the $V_2$ water loss term was calibrated while the modified SMAR was not calibrated at all. This gives us insights on the performance of the original and modified SMAR model against our reference SMN, e.g. how the SMAR model performs on profile/plot scale and if the use on the entire SMN already provides results with lower accuracy. This provides an additional benchmark for the evaluation of the results which is more robust than the arbitrary
threshold of 0.06 cm³/cm³ which has been used in previous studies to evaluate the SMAR performance.

The CRNS-derived soil moisture time series always represents a weighted soil moisture average because by the nature of the geophysical method, it is more sensitive to soil moisture changes close to the neutron detector. When the performance (goodness of fit) of the CRNS-derived surface soil moisture time series is assessed, it should always show a better performance
when the reference average soil moisture derived from e.g. the SMN is weighted according to the sensitivity of CRNS. Consequently, the assessment of the different neutron-to-soil moisture transfer functions is done accordingly and our analyses underline the improved performance when the reference soil moisture from the SMN is weighted.
However, this indeed means that a (CRNS-derived) weighted field-scale soil moisture average is used as the input for the SMAR model while we compare the modelled second-layer soil moisture time series with arithmetic averages of the SMN in
the second layer.
To gain further understanding how the weighting procedure of the SMN affects the depth-extrapolated soil moisture time series, we added two additional depth-extrapolation scenarios: Using the arithmetic and weighted averages surface soil moisture from the SMN ($SMN_{arithmetic}$ and $SMN_{weighted}$) as the input for the different extrapolation approaches. The difference in the performance between $SMN_{arithmetic}$ and $SMN_{weighted}$ then allows to draw conclusion on the impact of the weighting
procedure on the goodness of fit.

**R1-2:** First, I found misleading the arguments to support the use of SMAR. The Authors argue several times that "most approaches require a site-specific calibration using depth-profiles of in-situ soil moisture data, which are often not available." (L6-7). For this reason, you use SMAR. However, this approach also requires calibration. So your arguments do not support
your decision. Similarly, it is stated (L9) that "SMAR is usually also calibrated to sensor data, but could be applied without calibration if all its parameters were known" and L90. "This method does not require calibration if the environmental parameters are known." Well this is valid for any models and not only for SMAR. The problem is that it is often the case that we do not know the parameters, and especially at the scale of model application when soil is under investigation. The results of this study also confirm this statement. So your arguments do not support your decision to use SMAR as many other approaches have similar settings. Overall, I'm fine with the decision to test SMAR (L86). But I'm against the argument that this is physically based and it does not require adjustment in comparison to other approaches. The results of the present study also confirm my doubt. Comparison to other approaches, e.g., the exponential filter (Franz et al., 2020; Wagner et al., 1999) could strengthen this study.

Finally, the Authors tested several SMAR settings. Despite the overall exercises are remarkable, I'm wondering why testing approaches that already sound not appropriate. I.e., I found the assumptions behind the original SMAR model very arguable. E.g., a constant flux V2 (P and ET) is already debated when applied for estimating monthly groundwater recharge and it should be considered unrealistic over a long time series (years) for hourly to daily resolutions. The results show how V2 moves from, e.g., 50 to 300 mm h-1 in Table 3. These results are not consistent and support my doubt of using standard approach. So I'm not surprised if the Authors tried to modify the approach and I would remove the scenario with the constant V2. Similarly, the CRNS effective depth seems to be easily estimated (L279). So why testing a guess of 35 cm? I would remove this scenario. Overfitting a model calibrating all the parameters also seems to me a crash test and it does not seem to me a good approach for testing if a model is reliable. Several parameters are not in a physically consistent range and what was argued to be a physical approach has been destroyed. I would also remove this scenario.

**A1-2:** We agree that our formulations may have been partly misleading and inconsistent. The original SMAR model introduced by Manfreda et al. (2014) comprises different parameters which have a direct physical meaning such as the porosity, soil moisture at wilting point and field capacity. The $V_2$ parameter also has a physical meaning as it represents a water loss term comprising evapotranspiration and percolation losses from the second soil layer. Given the physical meaning of all parameters in the SMAR model, they are easier to be estimated compared to bulk fitting or calibration parameters. For this reason, we stated that the SMAR model allows for the estimation of soil moisture in a second, deeper soil layer without the necessity of in-situ reference information for calibration. Yet, we acknowledge that despite its physical meaning, the $V_2$ parameter is difficult estimate and is usually calibrated.

The reviewer mentions the soil water index which is often referred to as the exponential filter (Wagner et al. 1999). This very simple approach has been applied in numerous studies and only includes a single, bulk fitting parameter, namely the characteristic time length T. With its calibration against in-situ reference measurements, many processes are tried to be accounted for. Estimating T for a specific study site, e.g. from parameters such as soil depth, soil physical parameters proofed difficult (e.g. Zhang et al. 2017, Wang et al. 2017). This makes the use of the exponential difficult when no in-situ reference soil moisture measurements for calibrating T are available.

However, we agree to the reviewer that the additional testing of the exponential filter approach would strengthen the outcome of this study. Consequently, we added the depth-extrapolation with the exponential filter method to our analyses, repeated all scenarios also with the exponential filter method and added these scenarios to the revised manuscript.

As the reviewer points out, when the V2 parameter is calibrated against in-situ reference measurements, its absolute calibrated value increases as the second layer thickness increases. This is related to the nature of the SMAR model being a version of a storage model. For example, when the field capacity of the second layer is exceeded by 10 cm$^3$/cm$^3$ or 10 vol.-%, the entire 10 vol.-% percolate from the second layer to larger depths. Expressed in mm, this value is higher for a second soil layer with a thickness of 1000 mm compared to 500 mm. This explains higher values for $V_2$ in units of mm when the thickness of the second soil layer increases.

Nevertheless, we do agree that the constant $V_2$ is problematic but in our opinion keeping this scenario in the manuscript is important to compare our introduced modified SMAR (estimating a dynamic $V_2$) with the original SMAR (calibrating a constant $V_2$).

The reviewer also argues that some scenarios (first layer depth of 35 cm) and fitting all soil physical parameters should be removed. We are thankful about this comment as we think that this will help to streamline the story of our manuscript, to make it more digestible for the reader and to highlight the key points of this study. We removed all scenarios where soil physical parameters are calibrated (except $V_2$). We also removed the scenarios for hourly time steps and only keep scenarios with daily time steps as this is the temporal resolution for which both, the SMAR model and the exponential filter have been designed for.

**R1-3:** L3-5. deeper than what? I would rephrase otherwise it is not clear to what you refer when saying "deeper", e.g., moving the sentence "Many of these applications require information on soil water dynamics in deeper soil layers." After " Cosmic-ray neutron sensing (CRNS) allows for non-invasive monitoring of field-scale soil moisture across several hectares around the instrument but only for the first few tens of centimeters of the soil."

**A1-3:** We rearranged the sentences accordingly:

Original: "Many of these applications require information on soil water dynamics in deeper soil layers. Cosmic-ray neutron sensing (CRNS) allows for non-invasive monitoring of field-scale soil moisture across several hectares around the instrument but only for the first few tens of centimeters of the soil."

Modification: "Cosmic-ray neutron sensing (CRNS) allows for non-invasive monitoring of field-scale soil moisture across several hectares around the instrument but only for the first few tens of centimeters of the soil. Many of these applications require information on soil water dynamics in deeper soil layers."

**R1-4:** L8. Personally, I'm not too much on the philosophical discussion about process-based or empirical approaches (e.g., (Hrachowitz and Clark, 2017)). But to call a bucket model with several empirical assumptions a physical-based approach could be highly criticized. Please consider rewording.

**A1-4:** We rephrased the sentence in the following way:

Original: "The physically-based soil moisture analytical relationship SMAR is usually also calibrated to sensor data, but could be applied without calibration if all its parameters were known. However, in particular its water loss parameter is difficult to estimate."

Modification: "The soil moisture analytical relationship SMAR is usually also calibrated to sensor data, but due to the physical meaning of each model parameter, it could be applied without calibration if all its parameters were known. However, in particular its water loss parameter is difficult to estimate."

**R1-5:** L36. Despite, I agree that establishing an extensive point sensor network requires a lot of effort, the international soil moisture network ISMN (Dorigo et al., 2021) provides a good example that it is worth citing.

**A1-5:** We added this reference to our manuscript:

Original: "As a consequence, extensive point sensor networks which allow for the estimation of field-scale soil moisture are often restricted to a rather small number of research related monitoring sites such as the Terrestrial Environmental Observatories (TERENO, www.tereno.net) in Germany (e.g., Zacharias et al., 2011; Bogena et al., 2018; Kiese et al., 2018; Heinrich et al., 2018)."

Modification: "As a consequence, extensive point sensor networks which allow for the estimation of field-scale soil moisture are often restricted to a rather small number of research related monitoring sites such as the Terrestrial Environmental Observatories (TERENO, www.tereno.net) in Germany (e.g., Zacharias et al., 2011; Bogena et al., 2018; Kiese et al., 2018; Heinrich et al., 2018) or the International Soil Moisture Monitoring Network (ISMN, Dorigo et al. 2021) covering sites around the globe."

**R1-6:** "L80 and L82. To my knowledge, (Wagner et al., 1999) proposed the use of an exponential filter approach to estimate soil water content at deeper soil layers based on surface soil moisture. Instead the term SWI refers to a quantity of water content between 0-100%. Thus, I would not call the extrapolation approach "soil water index" but rather "exponential filter approach".

**A1-6:** We exchanged/added the term "soil water index" with "exponential filter approach" throughout a revised version of this manuscript.

**R1-7:** L128. How field capacity and wilting point have been estimated is missing and should be added.

**A1-7:** This information is already given in the caption of Tab. 1: "[...]. The soil moisture content at field capacity and wilting point were taken from tabulated values in Sponagel et al. (2005) according to the respecitve soil grain size class (medium-fine sand) and the soil bulk density of the individual layers."

**R1-8:** L134. The use of the manufacture's calibration function could be another source of errors that should be discussed in the result section.

**A1-8:** This is correct. Adding the additional analyses and the new results requested by reviewer #1 in R1-1 and R1-2 will lead to severe changes in content and structure of the manuscript. We added the following sentence to chapter 3.4:

"When assessing the results obtained from using the in-situ sensor, it should also be noted that the use of the manufacturer's calibration function adds additional uncertainty to the results."

**R1-9:** L154. The fact that you use a 25h moving average could be a reason to not be able to capture fast soil moisture changes? This could be discussed in the result section.

**A1-9:** The smoothing of the corrected CRNS neutron signal prior transferring it soil moisture may have some effect on capturing fast soil moisture changes. Although we think that this effect is not responsible for the poor performance of SMAR, we added the following sentence to chapter 3.4:

"Although this effect may be small, particularly on the daily time step, smoothing hourly CRNS data prior estimating surface soil moisture and aggregating daily soil moisture estimates could contribute to the poorer performance of depth-extrapolated time series in the CRNS scenarios compared to the SMN scenarios."

**R1-10:** Eq.5. I guess h is air humidity. Please clarify.

**A1-10:** This is correct. We added this information accordingly:

Original: "The universal transport solution (UTS), eq. (5) – eq. (6), (Köhli et al., 2021) accounts for the changing relationship between neutrons and soil moisture under different conditions of absolute humidity."

Modification: "The universal transport solution (UTS), eq. (5) – eq. (6), (Köhli et al., 2021) accounts for the changing relationship between neutrons and soil moisture under different conditions of absolute humidity $h$ in g m$^{-3}$."

**R1-11:** L179. Please clarify how look-up-table approach works. What are also the assumptions/approximations in using this approach?

**A1-11:** The look-up table approach is necessary because the UTS equation cannot be simply rearranged to be solved for soil moisture. Instead, one approach is to calculate the neutron intensity from the UTS for each soil moisture value in a range from 0.0001 to 0.5 cm³/cm³ in 0.0001 cm³/cm³ steps and compare the derived neutron intensity with the observed neutron intensity at each time step. The soil moisture value which produces the smallest absolute difference between the predicted neutron intensity and observed neutron intensity is then used for each time step. We added this information:

Original: "Soil moisture can be derived from the UTS using numerical inversion or a look-up table approach which is used in this study."

Modification: "Soil moisture can be derived from the UTS using numerical inversion or a look-up table approach which is
used in this study. In the look-up table approach, soil moisture values in the range from 0.0001 to 0.5 cm$^3$ cm$^{-3}$ in steps of 0.0001 cm$^3$ cm$^{-3}$ are used to predict the neutron intensity using the UTS for each time step. For each time step, the soil moisture value producing the smallest absolute difference between the observed and predicted neutron intensity is then assigned as the CRNS-derived soil moisture value."

**R1-12:** L181. So how Nd has been actually calculated? Is it related to N0? Please clarify.

**A1-12:** $N_D$ has been calculated just like the look-up table approach described in the response to the previous comment. For the hours of the calibration campaign in February 2019, soil moisture from independent soil samples and absolute humidity from local temperature and relative humidity measurements were used to predict a neutron intensity with the UTS for a range of $N_D$ values. The $N_D$ value with the smallest RMSE between observed and predicted neutron intensities was then used. The $N_D$ value needs to be calibrated for each version of the UTS (MNCP thl, MCNP drf, URANOS thl, URANOS drf).

$N_0$ is derived equally for the standard transfer function. However, the physical meaning of the two parameters differs and a direct, unique relationship between the calibration parameters does not exist as the absolute humidity term in the UTS changes this relationship. We will added the following sentence to line 201:

Original: "Reference soil moisture information from the soil sampling campaign in February 2019 was weighted accordingly and used for calibrating both transfer functions."

Modification: "Reference soil moisture information from the soil sampling campaign in February 2019 was weighted accordingly and used for calibrating both transfer functions. $N_0$ and $N_D$ were iteratively calibrated. For $N_0$, the value producing the smallest RMSE between soil moisture from soil samples and the one predicted with eq. 1-4 was chosen. For $N_D$ soil moisture information derived from soil samples for the hours of the sampling campaign were used to predict neutron intensities with eq. 5-6. The $N_D$ producing the smallest RMSE between predicted and observed neutron intensities was chosen."

**R1-13:** L187. What are the differences between URANOS type and MCNP type equation?

**A1-23:** The UTS is based on neutron transport simulations conducted with two different neutron transport model codes: URANOS and MCNP. As expected two different models produce slightly different results and hence, different fitting parameters for $p_1$-$p_{10}$ have been obtained. We tested all available parameter sets for the UTS to determine which should be used and our results are in line with the initial findings of Köhli et al. (2021) who introduced the UTS.

**R1-14:** L211. Are you sure you have to move back and forward from volumetric soil moisture and relative saturation? Could you not try directly expanding eq. 7 directly for volumetric soil moisture?

**A1-14:** In case of the SMAR model we have to stick to the procedures and units defined. This would be different for the exponential filter method which can be applied to soil moisture data in units of relative saturation and $cm^3/cm^3$ (Bouaziz et al. 2020). However, the calibrated characteristic time length parameter T is likely to be different for a soil moisture time series that have been previously transformed to relative saturation and one in units of $cm^3/cm^3$. This should be considered when comparing different studies reporting calibrated values for T.

**R1-15:** L248. I guess the estimation would be very sensitive to the length of the time series. Did you use the entire time series for estimating ET? Please clarify.

**A1-15:** The average $ET_2$ for the entire time series determined based on eq. 12-13 would indeed be dependent on the length of the surface soil moisture time series. However, we are not determining an average $ET_2$ and $P_2$ for the entire time series and then use the subsequently derived time series average $V_2$ to run the SMAR model for deriving soil moisture in the second layer.

Instead, as it can be seen in eq. 13, the $ET_2$ value is calculated and used individually for each time step and so it also done for $P_2$. This also means that each time step of in the modified SMAR model receives its individual $ET_2$ and $P_2$ value and hence, also and individual varying $V_2$ water loss. We added the following modification:

Original: "Here, we estimate the amount of evapotranspiration from the deeper layer $ET_2$ based on the difference between the current and past value of relative saturation of the first layer, by scaling the value to the dimension (i.e. extent) of second layer and by considering the difference in cumulative root fraction between both layers, assuming that root water uptake for ET is larger in the layer with more roots eq. (13)."

Modification: "Here, we estimate the individual amount of evapotranspiration from the deeper layer $ET_2$ for each time step based on the difference between the current and past value of relative saturation of the first layer, by scaling the value to the dimension (i.e. extent) of second layer and by considering the difference in cumulative root fraction between both layers, assuming that root water uptake for ET is larger in the layer with more roots eq. (13)."

**R1-16:** L490. Well, the discovery that root water uptake varies with time and depth depending on the water availability goes probably back to the introduction of irrigation practices in the history. I guess Maysonnave et al.,2022 showed more than that. If the Authors remove the scenario with V2 constant, this comment could also be probably removed. Otherwise please rephrase.

**A1-16:** As stated in A1-1, we would like to keep the results for the original SMAR version in this manuscript to compare it with our modified SMAR version and, as requested by reviewer #1, also with the exponential filter in a revised version of our manuscript. We interpret R1-16 in the way that the reference of Maysonnave et al. (2022) is not appropriate here and that information on depth and water availability related root-water uptake is well established. Consequently, we removed this sentence.

**R1-17:** The Authors have stated that this study is the first who evaluates the UTS (L515). If this is the case I think is worth clarifying this from the beginning and provide some additional information to better understand how it works and is implemented.

**A1-17:** Yes, to our knowledge this is the first study apart from Köhli et al. (2021) who introduced the UTS which compares the standard transfer function after Desilets et al. (2010) and UTS in terms of their performance in deriving soil moisture from CRNS. More information on the calibration of ND and the look-up table approach will also be added (see A1-11, A1-12, A1-13). We replaced the following statement in the introduction:

Original: "In addition, we apply different neutron-to-soil moisture transfer functions available to derive the surface soil moisture time series. This is done to assess which transfer function performs best and if a better CRNS-derived surface soil moisture time series translates into better estimates of the depth-extrapolated soil moisture."

Modification: "Different approaches exist to derive soil moisture from observed neutron signals. The standard approach after Desilets et al. (2010) is commonly used to derive soil moisture from CRNS but has been found insufficient especially at observation sites with low soil moisture contents. New approaches include the interdependence of the relationship between neutrons and soil moisture (Köhli et al., 2021) and report an improved estimation of soil moisture with CRNS.

The three depth-extrapolation approaches (SMAR, modified SMAR and exponential filter) are therefore applied using different surface soil moisture time series, including single point-scale in-situ sensor profiles, averages of the entire in-situ sensor network and CRNS-derived soil moisture from different neutron-to-soil moisture transfer functions in order to investigate the performance of the different approaches and if a better CRNS-derived surface soil moisture time series translates into better estimates of the depth-extrapolated soil moisture."

**Additional references:**

Bouaziz, L. J. E., Steele-Dunne, S. C., Schellekens, J., Weerts, A. H., Stam, J., Sprokkereef, E., Winsemius, H. H. C., Savenije, H. H. G., and Hrachowitz, M.: Improved Understanding of the Link Between Catchment-Scale Vegetation Accessible Storage and Satellite-Derived Soil Water Index, Water Resources Research, 56, e2019WR026365, https://doi.org/10.1029/2019WR026365, 2020.

Dorigo, W., Himmelbauer, I., Aberer, D., Schremmer, L., Petrakovic, I., Zappa, L., Preimesberger, W., Xaver, A., Annor, F., Ardö, J., Baldocchi, D., Bitelli, M., Blöschl, G., Bogena, H., Brocca, L., Calvet, J.-C., Camarero, J.J., Capello, G., Choi, M., Cosh, M.C., van de Giesen, N., Hajdu, I., Ikonen, J., Jensen, K.H., Kanniah, K.D., de Kat, I., Kirchengast, G., Kumar Rai, P., Kyrouac, J., Larson, K., Liu, S., Loew, A., Moghaddam, M., Martínez Fernández, J., Mattar Bader, C., Morbidelli, R., Musial, J.P., Osenga, E., Palecki, M.A., Pellarin, T., Petropoulos, G.P., Pfeil, I., Powers, J., Robock, A., Rüdiger, C., Rummel, U., Strobel, M., Su, Z., Sullivan, R., Tagesson, T., Varlagin, A., Vreugdenhil, M., Walker, J., Wen, J., Wenger, F., Wigneron, J.P., Woods, M., Yang, K., Zeng, Y., Zhang, X., Zreda, M., Dietrich, S., Gruber, A., van Oevelen, P., Wagner, W., Scipal, K., Drusch, M., and Sabia, R.: The International Soil Moisture Network: serving Earth system science for over a decade. Hydrology and Earth System Sciences 25, 5749–5804. https://doi.org/10.5194/hess-25-5749-2021, 2021.

Wang, T., Franz, T. E., You, J., Shulski, M. D., and Ray, C.: Evaluating controls of soil properties and climatic conditions on the use of an exponential filter for converting near surface to root zone soil moisture contents, Journal of Hydrology, 548, 683-696, http://dx.doi.org/10.1016/j.jhydrol.2017.03.055, 2017.

We thank reviewer #2 for taking the time to review our manuscript and the helpful comments which will improve the quality of the manuscript.

**R2-1:** MAJOR: A large number of simulations have been carried out. It is difficult to make a summary of the results obtained and to clearly understand the more important scientific understanding that has been gained by carrying out the study. For example, when looking at the figures, it seems that no configuration gives satisfactory results, especially in terms of soil moisture dynamics. A RMSE lower than 0.06 cm^3/cm^3 is not very representative and depends on the dynamic range of the soil moisture data. I would strongly suggest reducing the number of simulations and focusing on more relevant results.

**A2-1:** We are thankful about this comment and this leads into the same direction compared to what reviewer #1 mentioned in his/her comments. As also suggested by reviewer #1, we removed the scenarios with hourly data, a first layer depth of 35 cm and all scenarios in which we calibrated all soil physical parameters. This helped to streamline our analyses and made the main points in our manuscript clearer.

**R2-1:** MAJOR: Why are the results not as good as expected? Does it depend on the SMAR model, or on the CRNS data, or on the soil moisture benchmark? This should be clarified. For example, a simulation with surface and deeper layers from the Soil Moisture Benchmark data can give insight into the performance of SMAR. Some additional analysis in this direction should be done.

**A2-2:** This is again a very similar comment to what has been mentioned by reviewer #1. We are glad to see that the improvements which need to be made on this manuscript are clear.

In addition to removing a large part of the scenarios as mentioned in A2-1, we performed the remaining scenarios of SMAR also based on the in-situ point-scale soil moisture sensors, only. This means that we used the surface soil moisture time series of the in-situ soil moisture sensors and applied all SMAR scenarios for two selected soil moisture sensor profiles. Furthermore, we then performed the analyses based on arithmetic and weighted averages from the entire soil moisture sensor network.

This gives us a better, site-specific benchmark to evaluate the original and modified version of the SMAR model by testing it on in-situ profile scale and in-situ network scale before testing the influence of a CRNS-derived soil moisture time series as the input. This way, we are also able to get more clues on the performance of the original SMAR model and our modified version.

**R2-3:** MODERATE: Other methods have been proposed to extrapolate soil moisture data from the surface to deeper layers. The exponential filter is the most commonly used approach. I would suggest a comparison with such an approach, again to provide additional information on the performance of the SMAR model.

**A2-3:** The reviewer is right. The exponential filter method (also referred to as the Soil Moisture Index) is a famous and often used approach to derive soil moisture time series in deeper layers from a surface soil moisture time series. We initially decided not to add this approach to our manuscript as its only bulk fitting parameter T is not directly linked to a single specific physical quantity and is difficult to predict for individual observation sites (e.g. Zhang et al. 2017, Wang et al. 2017). Hence, it always requires calibration against reference soil moisture information in the depth of interest. However, finding an approach which can be used without reference information in the depth of interest was the initial objective of this study.

Nevertheless, as also pointed out by reviewer #1, the exponential filter is a standard approach and comparing the original SMAR with calibrated water loss parameter $V_2$ and the completely uncalibrated modified SMAR with the exponential filter with calibrated T parameter can provide some additional insights into the performance of either SMAR version at our study site.

Following the response to the comments of reviewer #1, we repeated all analyses also with the exponential filter and added the results to a modified version of our manuscript.

**Additional references:**

[revised manuscript text omitted]